

# A taxonomic revision of *Garcinia* section *Xanthochymus* (Clusiaceae) in Thailand

Chatchai Ngernsaengsaruay[1,2], Pichet Chanton[1], Minta Chaiprasongsuk[1] and Nisa Leksungnoen[3]

[1] Department of Botany, Faculty of Science, Kasetsart University, Chatuchak, Bangkok, Thailand
[2] Biodiversity Center Kasetsart University (BDCKU), Chatuchak, Bangkok, Thailand
[3] Department of Forest Biology, Faculty of Forestry, Kasetsart University, Chatuchak, Bangkok, Thailand

## ABSTRACT

*Garcinia* section *Xanthochymus* (Clusiaceae) is revised for Thailand with four native species, *i.e.*, *G. dulcis*, *G. nervosa*, *G. prainiana*, and *G. xanthochymus*. All species are described with updated morphological descriptions, illustrations, and an identification key, together with notes on distributions, distribution maps, habitats and ecology, phenology, conservation assessments, etymology, vernacular names, uses, and specimens examined. Four taxa, *G. andamanica*, *G. andamanica* var. *pubescens*, *G. cambodgiensis* and *G. vilersiana*, are synonymized under *G. dulcis*, and two taxa, *G. nervosa* var. *pubescens* and *G. spectabilis*, are newly synonymized under *G. nervosa*. Nine names are lectotypified: *G. dulcis* and its associated synonyms (*G. cambodgiensis* and *G. vilersiana*), *G. nervosa* and its associated synonyms (*G. andersonii*, *G. nervosa* var. *pubescens*, and *G. spectabilis*), *G. prainiana*, and *G. xanthochymus*. All species have a conservation assessment of Least Concern (LC). The fruits of all species are edible and have a sour or sweet-sour taste.

# INTRODUCTION

*Garcinia* L. is a group of evergreen trees, occasionally shrubs, which are usually dioecious, but sometimes polygamo-dioecious (also called trioecious). It also has some obligately and facultatively agamospermous species and is the largest genus in the Clusiaceae Lindl. (Guttiferae Juss.). The genus consists of 405 accepted species, and is distributed throughout the tropics and subtropics (*POWO, 2023*) with centers of diversity in Southeast Asia and Madagascar (*Sweeney & Rogers, 2008*). In Asia, *Garcinia* is most diverse in the Malesian region but also spreads north into southern China, west to India, and east to the Micronesian islands (*Nazre et al., 2018*). Previous studies on *Garcinia* revealed that the Indian subcontinent (including India, Andaman and Nicobar Islands, Nepal, Bhutan, Bangladesh, and Sri Lanka) has 45 species (*Anderson, 1874*; *King, 1890*; *Maheshwari, 1964*; *Kostermans, 1980*; *Long, 1984*; *Singh, 1993*; *Srivastava, 1994*; *Pathirana & Herat, 2004*; *Nimanthika & Kaththriarachchi, 2010*; *Begum, Barthakur & Sarma, 2013*; *Sabu et al., 2013*; *Dutta et al., 2014*; *Tabassum, 2015*; *Sarma, Shameer & Mohanan, 2016*; *Shameer, Sabu & Mohanan, 2017*; *Shameer et al., 2021*), Myanmar has 19 species

Corresponding author
Chatchai Ngernsaengsaruay, fsciccn@ku.ac.th

(*Anderson, 1874*; *Kurz, 1874*; *Kurz, 1877*; *Maheshwari, 1964*; *Singh, 1993*; *Nazre et al., 2018*; *Sweeney, Nwe & Armstrong, 2022*), China has 20 species (*Li et al., 2007*), Indo-China (Vietnam, Laos, and Cambodia) has 31 species (*Pitard, 1910*; *Gagnepain, 1943*; *Hô, 1991*; *Toyama et al., 2017*; *Tagane et al., 2018*; *Tuan et al., 2023*), the Malesian region has 82 species, Peninsular Malaysia has 48 species (*King, 1890*; *Ridley, 1922*; *Merrill, 1923*; *Backer & Bakhuizen van den Brink, 1963*; *Chin, 1973*; *Kochummen & Whitmore, 1973*; *Whitmore, 1973*; *Turner, 1995*; *Nazre et al., 2018*), and Australia has 12 species (*Cooper, 2013*).

In Thailand, the genus *Garcinia* was enumerated by *Craib (1925)*, with 20 species. *Gardner, Sidisunthorn & Anusarnsunthorn (2000)* listed six species from the northern region and *Gardner, Sidisunthorn & Chayamarit (2015)* recorded 23 species (including five unidentified species) from the peninsular region. More recently, *Ngernsaengsaruay & Suddee (2016)* and *Ngernsaengsaruay & Suddee (2022)* described two new species: *G. nuntasaenii* Ngerns. & Suddee from north-eastern and *G. santisukiana* Ngerns. & Suddee from eastern Thailand, respectively. *Ngernsaengsaruay (2022)* reported three species in *Garcinia* sect. *Brindonia* in Thailand: *G. atroviridis* Griff. ex T. Anderson, *G. lanceifolia* Roxb., and *G. pedunculata* Roxb. ex Buch.-Ham. *Ngernsaengsaruay, Duangnamon & Boonthasak (2022)* and *Ngernsaengsaruay et al. (2023)* published additional new records from Peninsular Thailand: *G. dumosa* King and *G. exigua* Nazre, respectively. Finally, *Ngernsaengsaruay et al. (2022)* described *G. siripatanadilokii* Ngerns., Meeprom, Boonthasak, Chamch. & Sinbumr. as a new species from Peninsular Thailand. From these publications, the genus has a total of c. 28 accepted species or more in Thailand.

*Garcinia* is characterized by a dioecious habit (sometimes apparently polygamo-dioecious); yellow, pale yellow, white, cream, or clear latex secreted from cut boles, twigs, leaves, and fruits; terminal buds concealed between the bases of the uppermost pair of petioles; decussate leaves with scattered black or brown gland dots, or interrupted wavy lines of differing lengths; male flowers with many to numerous stamens untied into a column in the center of the flower, into a variously lobed or angled, or into 4 or 5 separate bundles; berry fruits and seeds usually with thick or thin fleshy pulp (*Ngernsaengsaruay, Duangnamon & Boonthasak, 2022*).

The latest monograph for the genus *Garcinia* was published by *Vesque (1893)*, who recognized three subgenera and nine sections based on floral morphology. Among the sections was *G.* sect. *Xanthochymus* with 19 species. The most recent worldwide sectional treatment of *Garcinia* was provided by *Jones (1980)* in an unpublished Ph.D. thesis. She classified the genus into 14 sections based chiefly on floral morphology. She maintained *G.* sect. *Xanthochymus* as a separate section with 42 species. The section is widely distributed from Tropical Africa, Madagascar, the Indian subcontinent, southern China, and throughout Southeast Asia to Australia. It is distinguished by the combination of usually five-merous flowers (rarely four-merous) that have nectaries occupying the center of the male flowers ("disks") or positioned beneath the ovary of female flowers ("rings" or "appendages") and male flowers with stamens united into bundles with filaments united for at least 1/2 of their length; fleshy, thin-skinned fruits that are usually sinuously wrinkled when dry (*Anderson, 1874*; *Vesque, 1893*;

*Kochummen & Whitmore, 1973*; *Jones, 1980*). Also, *Jones (1980)* reports that the pollen is five- to seven-colporate and psilate. Several species are well known because they have edible fruits. In Thailand, six species have been recorded in *G.* sect. *Xanthochymus*: *G. cambodgiensis* Vesque, *G. dulcis* (Roxb.) Kurz, *G. nervosa* (Miq.) Miq., *G. prainiana* King, *G. vilersiana* Pierre, and *G. xanthochymus* Hook. f. ex T. Anderson (*Craib, 1925*; *Gardner, Sidisunthorn & Anusarnsunthorn, 2000*; *Office of the Forest Herbarium, Forest and Plant Conservation Research Office, Department of National Parks, Wildlife and Plant Conservation, 2014*; *Gardner, Sidisunthorn & Chayamarit, 2015*) and in Indo-China, four species have been recorded: *C. cambodgiensis*, *G. phuongmaiensis* V. S. Dang, H. Toyama & D. L. A. Tuan, *G. vilersiana*, and *G. xanthochymus* (*Pierre, 1882*; *Pierre, 1883*; *Pitard, 1910*; *Gagnepain, 1943*; *Hô, 1991*; *Newman et al., 2007*; *Tuan et al., 2023*). There are many species within the section which cannot easily be distinguished from their close relatives (*e.g.*, *G. cambodgiensis*, *G. dulcis*, *G. vilersiana*, *G. xanthochymus*). A taxonomic revision of the genus *Garcinia* in Thailand has recently been undertaken by the first author as part of the Flora of Thailand. However, identifications mostly rely on the literature, and this is the case for *G.* sect. *Xanthochymus*, which has never been revised for Thailand. Therefore, in this paper, we provide an updated account here in order to present a taxonomic treatment that includes synonymizations, lectotypifications, a key to the species, detailed morphological descriptions, illustrations, distributions, distribution maps, habitats and ecology, phenology, conservation assessments, etymology, vernacular names, uses, and specimens examined.

## MATERIALS & METHODS

The collected specimens were examined by consulting taxonomic literature (*e.g.*, *Anderson, 1874*; *Pitard, 1910*; *Ridley, 1922*; *Gagnepain, 1943*; *Maheshwari, 1964*; *Whitmore, 1973*; *Singh, 1993*), and by comparing with herbarium specimens deposited in the following herbaria (Aarhus University (AAU); Bangkok Herbarium, Plant Varieties Protection Office, Department of Agriculture (BK); The Forest Herbarium, Department of National Parks, Wildlife and Plant Conservation (BKF); The Natural History Museum (BM); University of Copenhagen (C); Chiang Mai University (CMUB); Royal Botanic Garden (K), Muséum National d'Histoire Naturelle (P); Prince of Songkla University, Hat Yai District, Songkla Province, Thailand (PSU); Queen Sirikit Botanic Garden Herbarium, The Botanical Garden Organization (QBG); National Parks Board Singapore (SING)) and those included in the digital herbarium databases of AAU (https://www.aubot.dk/search_form.php), AU (http://www.nsii.org.cn/2017/home-en.php), BM (https://www.nhm.ac.uk/our-science/collections/botany-collections.html), Meise Botanic Garden (BR) (http://www.botanicalcollections.be), CAL (https://ivh.bsi.gov.in/phanerogams), E (https://data.rbge.org.uk/search/herbarium/), G (http://www.ville-ge.ch/cjb/), K (including K-W) (http://www.kew.org/herbcat), L (including U) (https://bioportal.naturalis.nl/), P (https://science.mnhn.fr/institution/mnhn/collection/p/item/search), and the Department of Botany Collections, Smithsonian (US) (https://collections.nmnh.si.edu/search/botany/) (all herbaria acronyms follow *Thiers , 2023*, continuously updated). All specimens

cited have been seen by the authors unless stated otherwise. The taxonomic history of the species was compiled using the taxonomic literature and online databases (*IPNI, 2023*; *POWO, 2023*). The morphological characters, distributions, habitats, ecology, phenology, and uses were described from historic and newly collected herbarium specimens and the author's observations during field work. The vernacular names were compiled from the specimens examined and the literature (*e.g.*, *Office of the Forest Herbarium, Forest and Plant Conservation Research Office, Department of National Parks, Wildlife and Plant Conservation, 2014*). The assessment of conservation status was performed following the IUCN Red List Categories and Criteria (*IUCN Standards and Petitions Committee, 2022*) for a preliminary assessment of the conservation category in combination with GeoCAT analysis (*Bachman et al., 2011*) and field information. The calculation of Extent of Occurrence (EOO) and Area of Occupancy (AOO) are based on GeoCAT (https://www.kew.org/science/our-science/projects/geocat-geospatial-conservation-assessment-tool). We obtained permission to collect specimens from the Department of National Parks, Wildlife and Plant Conservation, DNP 0907.1/1593.

## RESULTS

### Taxonomic Treatment

**Garcinia L. sect. Xanthochymus** (Roxb.) Pierre, Fl. Forest. Cochinch. 1(5): 3. 1883; Vesque, Epharmosis 2: 14. 1889 et in A. DC. & C. DC., Monogr. Phan. 8: 254–255. 1893.—*Xanthochymus* Roxb., Pl. Coromandel 2(4): 51, t. 196. 1805. Type: *Xanthochymus pictorius* Roxb. = *Garcinia xanthochymus* Hook. f. ex T. Anderson.

*Tree* evergreen; latex usually white, sticky; branches decussate, horizontal; branchlets usually 4-ridged (except *G. prainiana*, terete), pubescent or glabrous. *Leaves* decussate, usually large and coriaceous, pubescent or glabrous; petiole usually transversely rugose. *Inflorescences* on short, leafless lateral branchlets (in axils of fallen leaves), sometimes terminal (*e.g.*, *G. prainiana*), cymose, in fascicles of several to many flowers, pubescent or glabrous. *Flowers* unisexual, sometimes bisexual, plants dioecious, sometimes polygamo-dioecious (*e.g.*, *G. dulcis*); bracteolate; sepals and petals quincuncial; sepals 5, unequal, margin ciliolate or eciliolate, pubescent or glabrous outside; petals 5, usually subequal, margin ciliolate or eciliolate. *Male flowers*: disk in the center of the flower, intrastaminal, 5-lobed, with lobes positioned between the stamen bundles and antesepalous (opposite sepals) (except *G. prainiana*, disk ring-shaped); stamens numerous, united into 5 bundles (phalanges), antepetalous (opposite petals); anthers very small; pistillode very small or absent. *Female flowers*: appendages (disk lobes) 5, antesepalous, alternating with staminode bundles; staminodes united into 5 bundles, antepetalous (except *G. prainiana*, appendages and staminodes absent); ovary 5-locular (except *G. prainiana*, 5–8-locular); stigma 5-lobed (except *G. prainiana*, weakly 5–8-lobed or indistinctly lobed). *Fruit* a fleshy berry, with sticky yellow latex, exocarp thin, usually sinuously wrinkled when dry.

A section of four species in Thailand; three species in Indo-China (*Garcinia dulcis*, *G. phuongmaiensis*, and *G. xanthochymus*).

## A key to the species of *Garcinia* sect. *Xanthochymus* in Thailand

1a. Inflorescences on short, leafless lateral branchlets; fully opened flowers with erect, concave petals and corolla forming a bowl shape, less than two cm in diam.; petals whitish pale green, creamish white or pale yellow; leaf base not subamplexicaul; petioles usually longer than six mm long; disk 5-lobed; fruits mostly subglobose, globose or broadly ovoid, with a short beak at the apex; persistent stigma deeply 5-lobed…………….…………………..…......……….....2

1b. Inflorescences mostly terminal; fully opened flowers with spreading petals, more than 2 cm in diam.; petals variable in color (pale yellow, yellowish pink, yellowish red, pinkish red, pink or red); leaf base often subamplexicaul; petioles short, up to six mm long; disk ring-shaped; fruits mostly depressed globose or depressed subglobose, slightly concave or flattened at the apex; persistent stigma circular, button-like...………...….………………………**3. G. prainiana**

2a. Leaves up to 32.5 cm long, thickly coriaceous, midrib raised as a prominent ridge and secondary veins raised on lower leaf surface; branchlets 4-ridged; pedicels terete…........………3

2b. Leaves more than 32.5 cm long, very thickly coriaceous, midrib and secondary veins strongly ridged on lower leaf surface; branchlets 4-ridged, two of these strongly ridged or narrowly winged; pedicels 4-angular…………....……………………………………**2. G. nervosa**

3a. Inflorescences in dense fascicles of flowers; leaves slightly bullate to bullate; branchlets, leaves, petioles, sepals, and pedicels mostly pubescent to glabrescent; plants polygamodioecious…………………………………………………………………………………**1. G. dulcis**

3b. Inflorescences in lax fascicles of flowers; leaves smooth (not bullate); branchlets, leaves, petioles, sepals, and pedicels glabrous; plants dioecious…………...……**4. G. xanthochymus**

**1. Garcinia dulcis** (Roxb.) Kurz, J. Asiat. Soc. Bengal, Pt. 2, Nat. Hist. 43(2): 88. 1874 et Forest Fl. Burma 1: 92. 1877; Pierre, Fl. Forest. Cochinch. 1(5): 4. 1883; Vesque, Epharmosis 2: 14. t. 87. 1889 et in A. DC. & C. DC., Monogr. Phan. 8: 312. 1893; King, J. Asiat. Soc. Bengal, Pt. 2, Nat. Hist. 59(2): 169. 1890; Koord. & Valeton, Bijdr. Boomsoort. Java 9: 359. 1903; Merr., Philipp. J. Sci. 3: 362. 1908; Ridl., Fl. Malay Penins. 1: 179. 1922; Merr., Enum. Philipp. Fl. Pl. 3: 84. 1923; Corner, Wayside Trees Mal. 1: 316. fig. 105. ed. 2. 1952; Backer & Bakh. f., Fl. Java 1: 386. 1963; Maheshw., Bull. Bot. Surv. India 6: 115. t. 1. fig. 4. 1964; Whitmore in Whitmore, Tree Fl. Malaya 2: 209. 1973; H. Keng, Concise Fl. Singapore: 48. 1990; E. W. M. Verheij & R. E. Coronel (eds), PROSEA 2: 175, 176. t. 176. 1992; N. P. Singh in B. D. Sharma & Sanjappa, Fl. Ind. 3: 109. 1993; S. Baruah et al., Ethnobot. Res. Appl. 21(33): 6. fig. 6. 2021; W. E. Cooper, Austrobaileya 9(1): 4. fig. 1A–1B. 2013; A. Begum, S. K. Barthakur & J. Sarma, Pleione 7(2): 546. t. 1. 2013; S. Gardner, P. Sidisunthorn & Chayam., Forest Trees S. Thailand 1: 352. fig. 541. 2015.—*Xanthochymus dulcis* Roxb. [Hort. Bengal.: 42. 1814, nom. nud.], Pl. Coromandel 3(3): 66. t. 270. 1820 et in Carey, Fl. Ind. 2: 631. 1832; Wight, Icon. Pl. Ind. Orient. 1(10): 10. t. 192. 1839.—*Garcinia elliptica* Choisy in DC., Prodr. 1: 561. 1824 [non *Garcinia elliptica* Wall., Numer. List. 4869. 1831, nom. nud.].—*Xanthochymus javanensis* Blume, Bijdr. Fl. Ned. Ind. 5: 216. 1825.—*Stalagmitis dulcis* Cambess., Mém. Mus. Hist. Nat. 16: 393, 426, 1828.—*Stalagmitis elliptica* G. Don, Gen. Hist. 1: 621. 1831.—*Stalagmitis javanensis* Spach, Hist. Nat. Vég. 5: 328. 1836. Type: Roxburgh's illustration, *Xanthochymus dulcis* Roxb., Pl. Coromandel 3(3): 66. t. 270 (*Roxburgh, 1820*) (lectotype designated here)

— *Garcinia vilersiana* Pierre, Bull. Mens. Soc. Linn. Paris 1: 348. 1882 et Fl. Forest. Cochinch. 1(5): t. 71B–C. 1883; Vesque, Epharmosis 2: 14. t. 85. 1889 et in A. DC. & C.

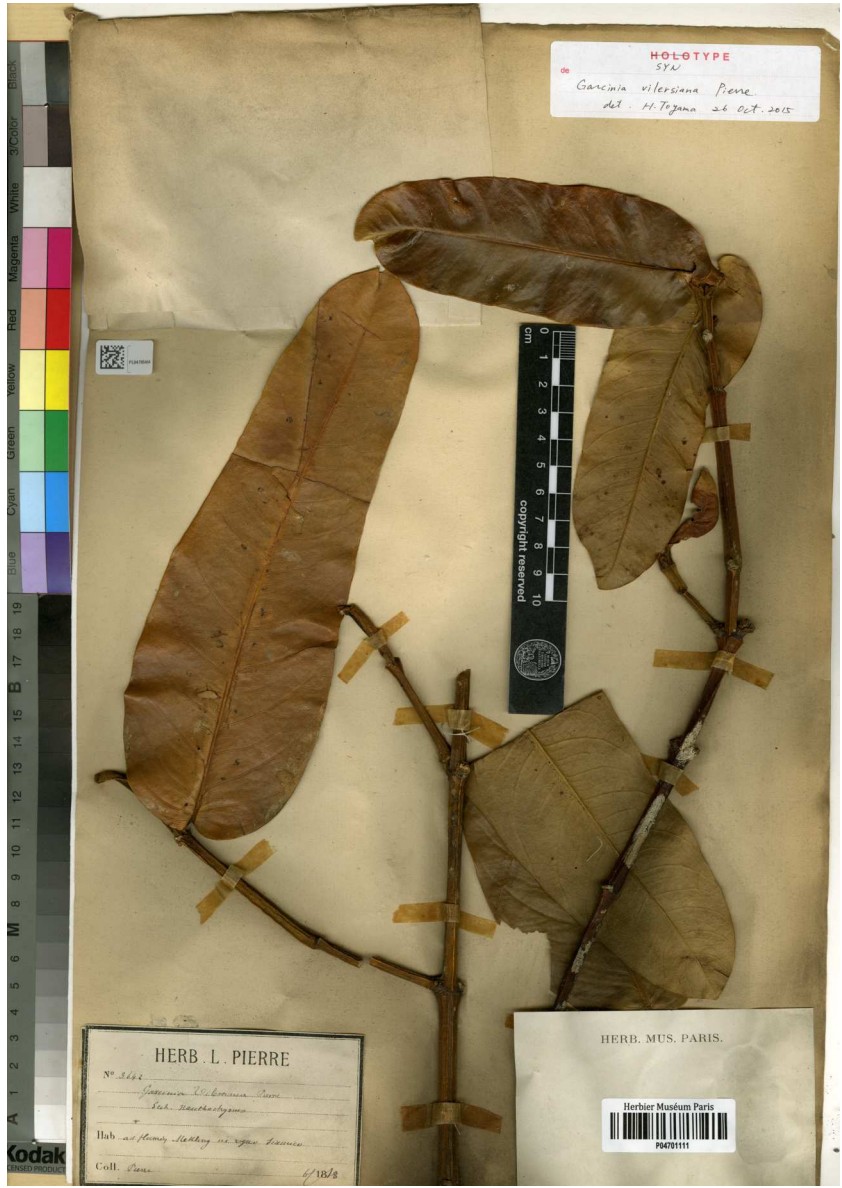

**Figure 1** Lectotype of *Garcinia vilersiana*, *Pierre 3642* (P [P04701111]) from Thailand, ad flumen Mekong in regno Siamico. Photo: MNHN - L. Randrihasipara - 2015, https://science.mnhn.fr/institution/mnhn/collection/p/item/p04701111. CC BY 4.0, https://creativecommons.org/licenses/by/4.0/.

DC., Monogr. Phan. 8: 318. 1893; Engl. in Engl. & Prantl, Die Naturlichen Pflanzenfamilien 3(6): 234. fig. 109D–F. 1895; Pit. in Lecomte et al., Fl. Indo-Chine 1(4): 297. fig. 29: 6–8. 1910; Craib, Fl. Siam. 1(1): 118. 1925; Gagnep. in Gagnep., Fl. Indo-Chine Suppl.: 257. 1943; Pételot, Arch. Rech. Agron. Cambodge Laos Vietnam 1: 64. 1952; P. H. Hô, Câyco Vietnam 1: 568. fig. 1572. 1991. Type: Thailand, ad flumen Mekong (also spelled Khong River) in regno Siamico, June 1868, *Pierre 3642* (lectotype selected here P [P04701111!]; isolectotypes P [P04701114!, P04701115!]), **syn. nov.** (Fig. 1).

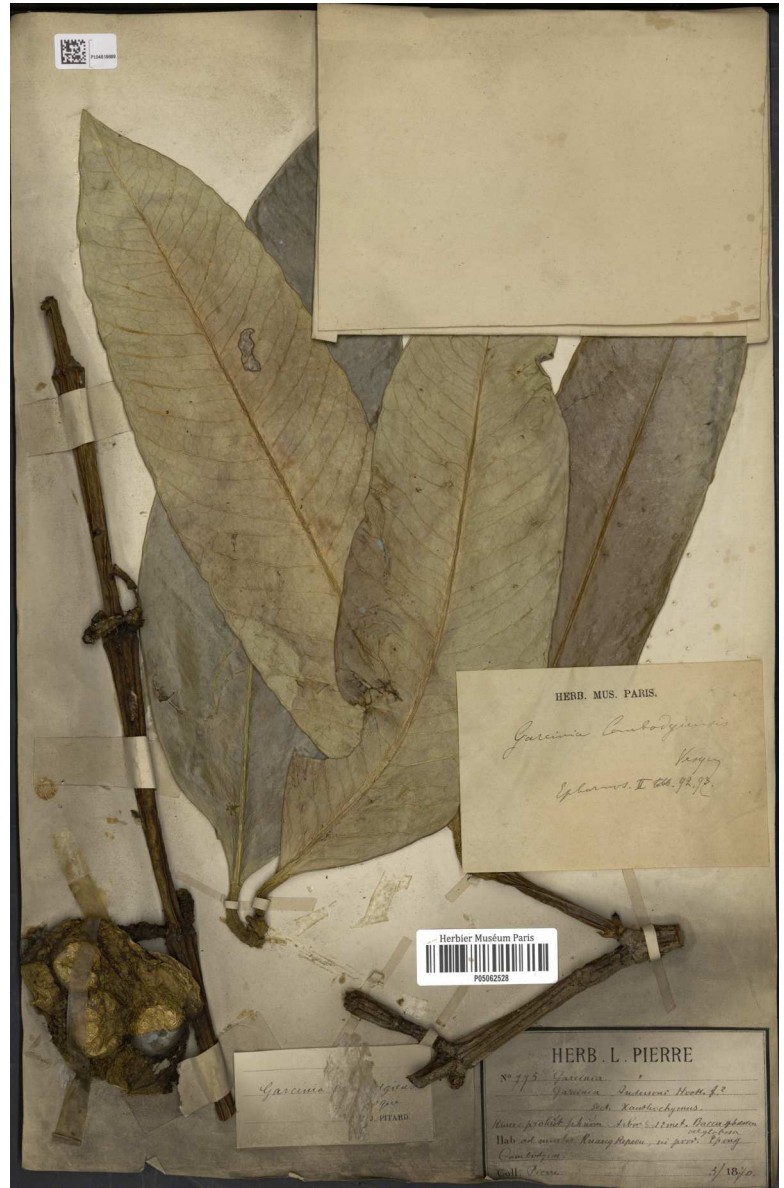

**Figure 2** Lectotype of *Garcinia cambodgiensis*, *Pierre 775* (P [P05062528]) from Cambodia, ad montes Kuang Repoeu, in prov. Tpong. Photo: Project: RENOBOTA, https://science.mnhn.fr/institution/mnhn/collection/p/item/p05062528. CC BY 4.0, https://creativecommons.org/licenses/by/4.0/.

—*Garcinia cambodgiensis* Vesque, Epharmosis 2: 14. t. 92, 93. 1889 et in A. DC. & C. DC., Monogr. Phan. 8: 316. 1893; Pit. in Lecomte et al., Fl. Indo-Chine 1(4): 298. 1910; Craib, Fl. Siam. 1(1): 114. 1925; Gagnep. in Gagnep., Fl. Indo-Chine Suppl.: 257. 1943.—*Garcinia andersoni* Pierre, Fl. Forest. Cochinch. 1(5): 1. t. 72. 1883. Type: Cambodia (originally "Cambodgiae" on the label), ad montes Kuang Repoeu, in prov. Tpong, May 1870, *Pierre 775* (lectotype selected here P [P05062528!]; isolectotypes K [K000677687!], P [P05062544, P05062564]), **syn. nov.** (Fig. 2).

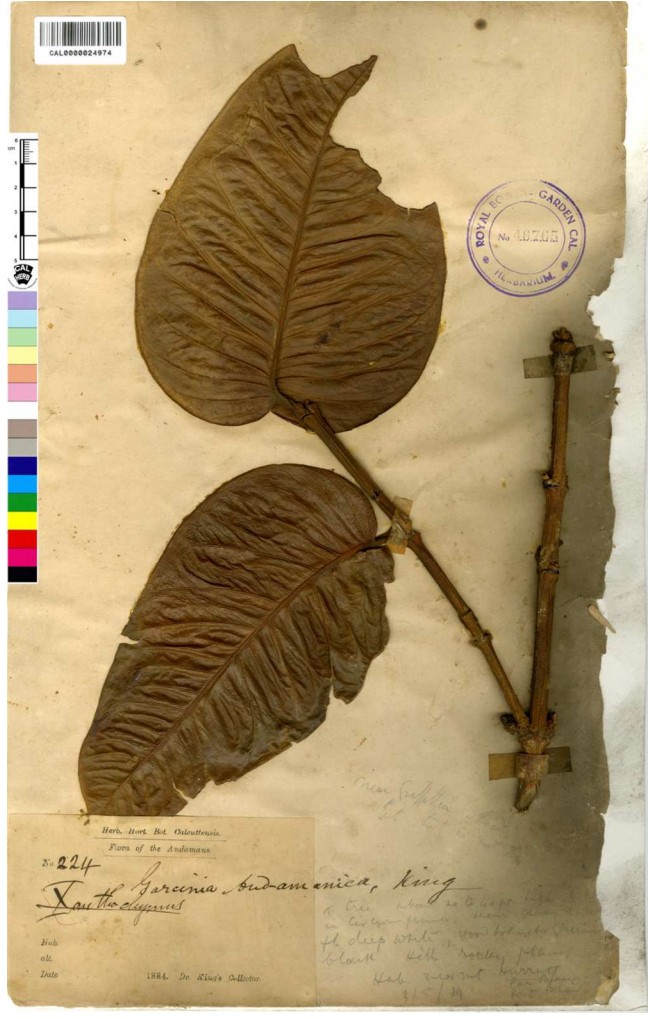

**Figure 3  Lectotype of *Garcinia andamanica*, King's Collector 224 (CAL [CAL0000024974]) from India, Andaman Islands, designated by *Shameer & Mohanan (2019)*.** Photo: ©The Director, Botanical Survey of India, Kolkata, https://ivh.bsi.gov.in/phanerogams-Details/en?link=CAL0000024973&column=szBarcode.

—*Garcinia andamanica* King, J. Asiat. Soc. Bengal, Pt. 2, Nat. Hist. 59(2): 170. 1890; Vesque in A. DC. & C. DC., Monogr. Phan. 8: 328. 1893; Brühl & King, Ann. Roy. Bot. Gard. (Calcutta) 5(2): 141. t. 169. 1896; Brandis, Indian Trees: 49. 1906; C. E. Parkinson, Forest Fl. Andaman Isl.: 89. 1923; Maheshw., Bull. Bot. Surv. India 6: 112. 1964; N. P. Singh in B. D. Sharma & Sanjappa, Fl. Ind. 3: 103. 1993; Shameer & N. Mohanan, Rheedea 29(2): 181. 2019. Type: India, Andaman Islands, 1884, *King's Collector 224* (lectotype designated by *Shameer & Mohanan (2019)*, CAL [CAL0000024974, photo seen], **syn. nov.** (Fig. 3).

—*Garcinia andamanica* King var. *pubescens* King, J. Asiat. Soc. Bengal, Pt. 2, Nat. Hist. 59(2): 170. 1890; Maheshw., Bull. Bot. Surv. India 6: 112. 1964; N. P. Singh in B. D. Sharma & Sanjappa, Fl. Ind. 3: 104. 1993. Type: India, Andaman Islands, 1884, *King's Collector*

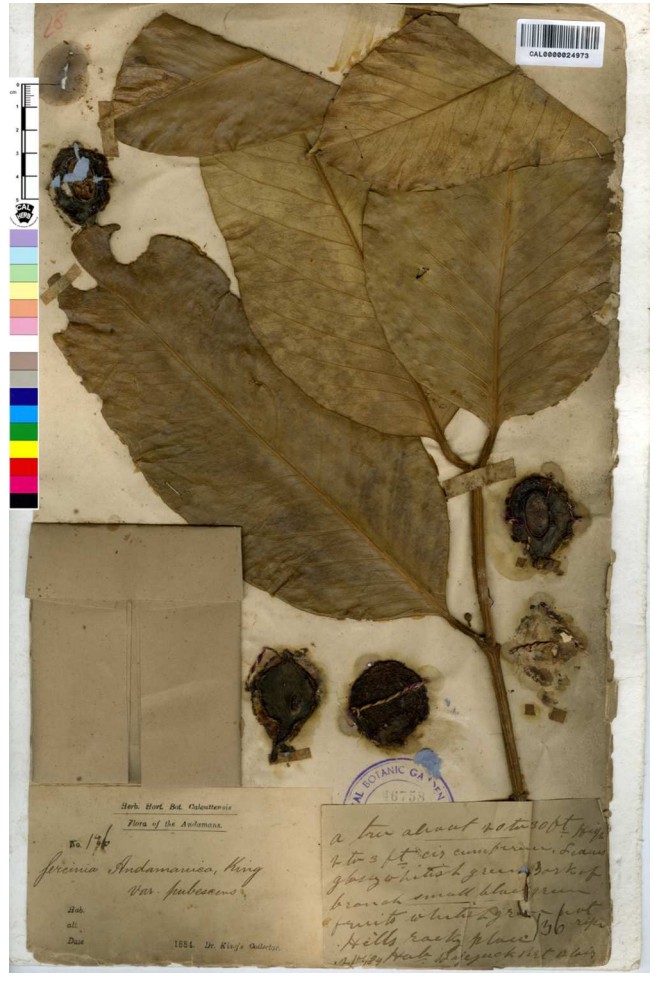

**Figure 4** Lectotype of *Garcinia andamanica* var. *pubescens*, *King's Collector 136* (CAL [CAL0000024973]) India, Andaman Islands, first step designated by *Maheshwari (1964)*.
Photo: ©The Director, Botanical Survey of India, Kolkata, https://ivh.bsi.gov.in/phanerogams-Details/en?link=CAL0000024973&column=szBarcode.

*136* (lectotype designated by *Maheshwari (1964)*, CAL [CAL0000024973, photo seen]; isolectotypes K [K000677630!], P [P05062496!]), **syn. nov.** (Fig. 4).

*Tree* evergreen, 5–20(–25) m tall, 30–160 cm girth, sometimes buttressed near the base of the stem in large trees; latex white, turning creamish white, sticky; branches decussate, horizontal; branchlets green, 4-ridged, pubescent, slightly pubescent or glabrescent, sometimes glabrous. *Bark* smooth or slightly rough, dark brown or greyish brown; inner bark pale yellow. *Terminal bud* concealed between the bases of the uppermost pair of petioles. *Leaves* decussate; lamina variable in shape, lanceolate-ovate or lanceolate (broadest at the basal part, gradually narrowing towards the apex), sometimes narrowly oblong, oblong or elliptic, 12.5–32.5 × 4.5–16 cm, apex acute or acuminate, sometimes obtuse, base obtuse, cuneate or subcordate, margin repand and slightly revolute, thickly coriaceous, slightly bullate or bullate, shiny dark green above, paler below, glabrous above,

pubescent, slightly pubescent or glabrescent, sometimes glabrous below (of lamina and veins), midrib and secondary veins flattened above, raised as a prominent ridge below, secondary veins 10–17 pairs, curving towards the margin connected in distinct loops and united into an intramarginal vein, conspicuous on both surfaces, with intersecondary veins, veinlets reticulate, visible on both surfaces, interrupted long wavy lines (glandular wavy lines, also called exudate containing canals) of differing lengths, running across the secondary veins to the apex or the margin, visible below; petiole green, 0.7–3.2 cm long, 2–6 mm in diam., not grooved, distinctly transversely rugose, indumentum same as in lamina, with a conspicuous basal appendage clasping the branchlet; young leaves shiny pale green; mature leaves turning greenish yellow to yellow before falling off. *Inflorescences* on short, leafless lateral branchlets (in axils of fallen leaves), cymose, in dense fascicles of 7–17 flowers. *Flowers* unisexual or bisexual, plants polygamo-dioecious, 5-merous, petals erect with overlapping edges and corolla forming a bowl shape; bracteolate; sepals and petals quincuncial, coriaceous, concave. *Male flowers* in fascicles of 11–17 flowers, 0.8–1.5 cm in diam.; bracteoles triangular, 0.7–2 × 0.6–2.7 mm, apex acute, pubescent; pedicel green, 0.5–1.7 cm long, 1–2 mm in diam., terete (circular in cross-section), pubescent or glabrescent, rarely glabrous; sepals 5, pale green, whitish pale green or green, broadly elliptic, semi-orbicular, suborbicular, 3–5.2 × 2.5–5 mm, unequal, apex rounded, margin ciliolate, pubescent or glabrescent, rarely glabrous outside; petals 5, whitish pale green, creamish white or pale yellow, broadly elliptic, elliptic or suborbicular, 5–8.5 × 4–8.5 mm, subequal, apex rounded, margin ciliolate; disk in the center of the flower, intrastaminal, yellow, 5-lobed, fleshy, pitted, lobes positioned between the stamen bundles, antesepalous; stamens numerous, united in 5 bundles, 6–13 in each bundle, antepetalous, 0.3–1 cm × 0.5–1 mm each bundle, creamish white or pale yellow; filaments 0.5–0.8 mm long; anthers yellow, 2 thecae, 0.2–0.4 × 0.3–0.7 mm; pistillode small or absent. *Female flowers* in fascicles of 7–9 flowers, 0.8–1.6 cm in diam.; bracteoles same as in male flowers; pedicel pale green or green, 1.2–2 cm long, widened at the apical part, middle part 1.6–2.7 mm in diam., apical part 2.7–4 mm in diam., terete, pubescent or glabrescent, rarely glabrous; sepals and petals same as or slightly larger than in male flowers; sepals 3–7.2 × 2.5–7.2 mm; petals 0.5–1 × 0.4–1 cm; appendages 5, antesepalous, alternating with staminode bundles, fleshy, pitted; staminodes united in 5 bundles, 2–5 in each bundle, antepetalous, 2.2–4.7 mm long each bundle, whitish pale green or creamish white; pistil 5–7.5 mm long; ovary pale green, subglobose, 3–4.5 × 4–6.5 mm, beaked, 0.5–2 × 1.4–2 mm (ovary including beaked looks like obpyriform in outline), unlobed, glabrous, 5-locular; stigma pale green, sessile, radiate, deeply 5-lobed, 3.5–5 mm in diam., papillate. *Bisexual flowers* same as in male and female flowers (androecium same as in male flowers; gynoecium same as in female flowers). *Fruits* berries, green, turning yellow or orangish yellow when ripe, smooth, glabrous, glossy, with sticky yellow latex, then exocarp becoming dark brownish black and sinuously wrinkled when dry, subglobose, globose or broadly ovoid, sometimes depressed globose, 3.8–7.5 × 4.5–8 cm, sometimes oblique, asymmetrical, unlobed, with a short, thick beak, pericarp 0.8–1.8 cm thick, exocarp thin; persistent stigma dark brown or blackish brown, radiate, deeply 5-lobed; persistent sepals green, slightly larger than in flowering materials; fruiting stalk green, 1.3–2.1 cm long, widened at the apical part, middle part 1.3–3.8 mm

in diam., apical part 2.5–4.5 mm in diam., pubescent or glabrescent, rarely glabrous. *Seeds* 1–5, sometimes aborted, brown mottled with irregular lines, ellipsoid, 1.7–3 × 0.9–2 cm, slightly oblique, rounded at both ends, with yellow or orangish yellow fleshy pulp (Fig. 5).

**Distribution.** India (Assam, Andaman Islands), Vietnam, Laos, Cambodia, Thailand, Peninsular Malaysia (also called Malaya) (Perlis, Perak, Kelantan), Indonesia [Java, Lesser Sunda Islands, Sulawesi, Moluccas (also called Maluku)], Borneo (Sabah), Philippines, New Guinea, Australia (Queensland), French Polynesia (Fig. 6).

**Distribution in Thailand.** Eastern: Buriram, Surin, Ubon Ratchathani; South-Western: Prachuap Khiri Khan; Central (cultivated); South-Eastern: Sa Kaeo, Prachin Buri, Chachoengsao, Chon Buri, Chanthaburi, Trat; Peninsular: Surat Thani, Krabi, Stun, Songkhla, Pattani, Narathiwat (Fig. 6).

**Habitat and Ecology.** It is found in dry evergreen, tropical lowland evergreen rain forests, and lower montane forests, often in limestone areas, sometimes along streams, 0–1,100 m alt.

**Phenology.** Flowering and fruiting more than once, nearly throughout the year; flowering usually in November to June; fruiting usually in January to July.

**Conservation Status.** *Garcinia dulcis* is widely distributed from India, Andaman Islands to North Queensland and Tahiti Islands. It is known from many localities and has a large Extent of Occurrence (EOO) of 14,325,860.58 km$^2$ and a relatively large Area of Occupancy (AOO) of 824 km$^2$. In Thailand, this species is known to be naturally distributed in the eastern, the south-western, the south-eastern and the peninsular regions, and has an EOO of 422,327.01 km$^2$ and an AOO of 184 km$^2$. Because of this wide distribution and the number of localities, it is considered Least Concern (LC).

**Etymology.** The specific epithet of *Garcinia dulcis* is a Latin word meaning sweet and refers to the ripe fruits that have a sweet-sour taste (*Stearn, 1992*; *Radcliffe-Smith, 1998*; *Gledhill, 2002*). The specific epithet of its new synonym, *G. vilersiana* is in honour of the late Charles Marie Le Myre de Vilers (1833–1918), a French naval officer, then departmental administrator. He was governor of the colony of Cochinchina (1879–1882). The specific epithet of its two new synonyms, *G. cambodgiensis* and *G. andamanica* are named after Cambodia and Andaman Islands, respectively where the type specimens for these species were collected.

**Vernacular Name.** Khai chorakhe (ไข่จระเข้) (Chanthaburi); Champhut (จำพูด) (Central); Taphut (ตะพูด) (Chanthaburi); Prahot (ปราโฮด), Prahut (ปราหูด) (Khmer-Surin); Pahut (ปะหูด) (Northeastern); Phahut (พะฮูด) (Buri Ram); Phawa bai yai (พะวาใบใหญ่) (Chanthaburi, Chonburi); Phut (พูด) (Satun); Maphut (มะพูด) (Central, Peninsular); Sompong (ส้มปอง), Sommuang (ส้มม่วง) (Chanthaburi); Baniti (Philippines); Cay vang nhua, Vang nhua (Vietnam); Madaw mu (Andaman Islands); Mundu (Javanese, Malay), Munu (Malay); Prahout (Cambodia); Yellow mangosteen (English).

**Uses.** *Garcinia dulcis* is often cultivated for its fruits. The fruits (pericarp and fleshy pulp) are edible and have a sour or sweet-sour taste. It is also grown in some botanical gardens as an ornamental tree to provide botanical education. It is cultivated as a fruit tree in Southeast Asia (*Allen, 1965*; *Begum, Barthakur & Sarma, 2013*). The sour fruits can be eaten raw or cooked, and they also make jams and preserves (*Corner, 1952*; *Sastri, 1956*;

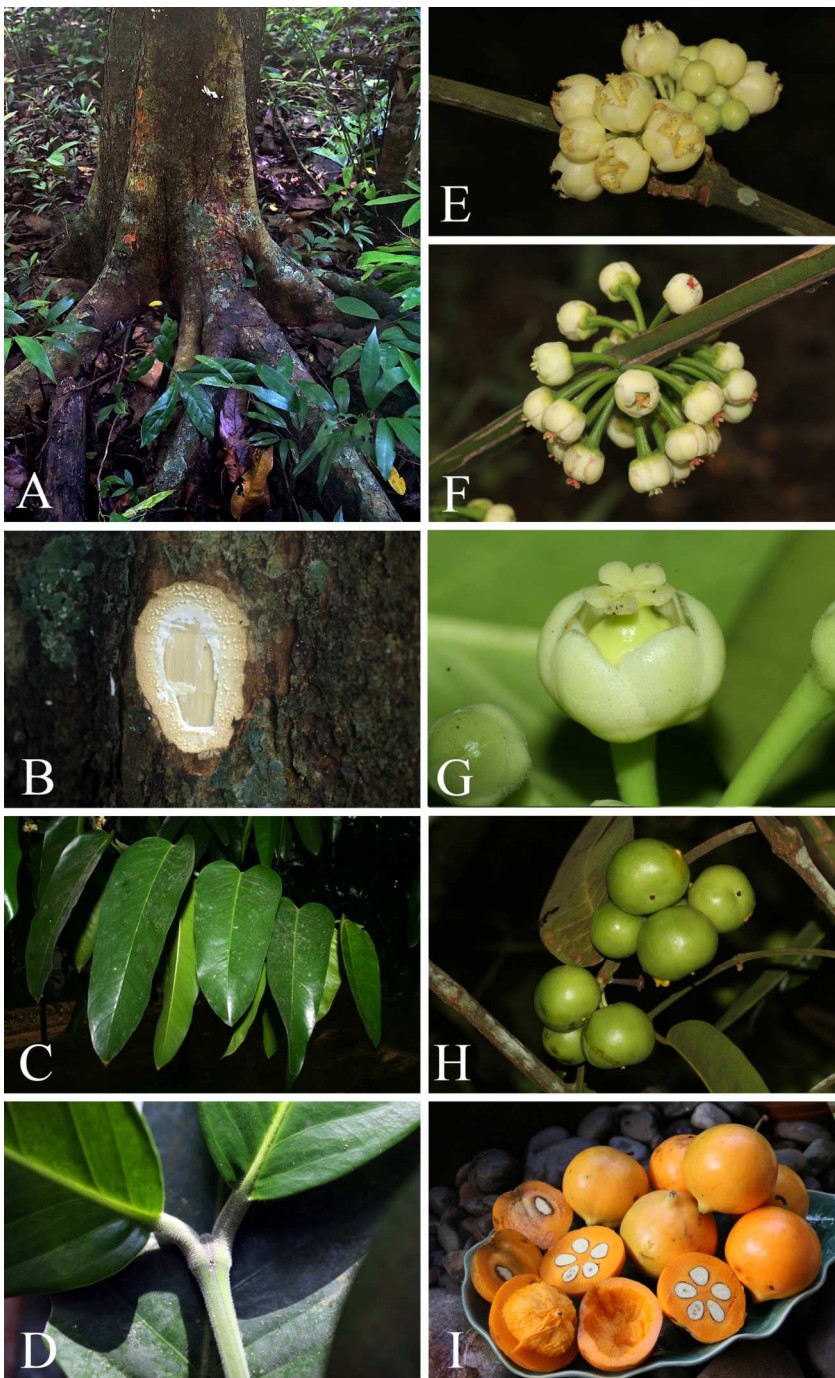

**Figure 5** *Garcinia dulcis.* (A) Trunk with buttressed base. (B) Outer bark, inner bark and slashed bark with white, turning creamish white latex. (C) Branchlet and leaves. (D) Terminal bud concealed between the bases of the uppermost pair of petioles. (E–F) Inflorescences on short, leafless lateral branchlets. (E) Inflorescence with fully opened male flowers and male flower buds. (F) Inflorescence with fully opened female flowers. (G) Fully opened female flowers. (H) Fruiting branchlets. (I) Ripe fruits, transverse and longitudinal sections of fruits with orangish yellow fleshy pulp and seeds. Photos: Chatchai Ngernsaengsaruay.

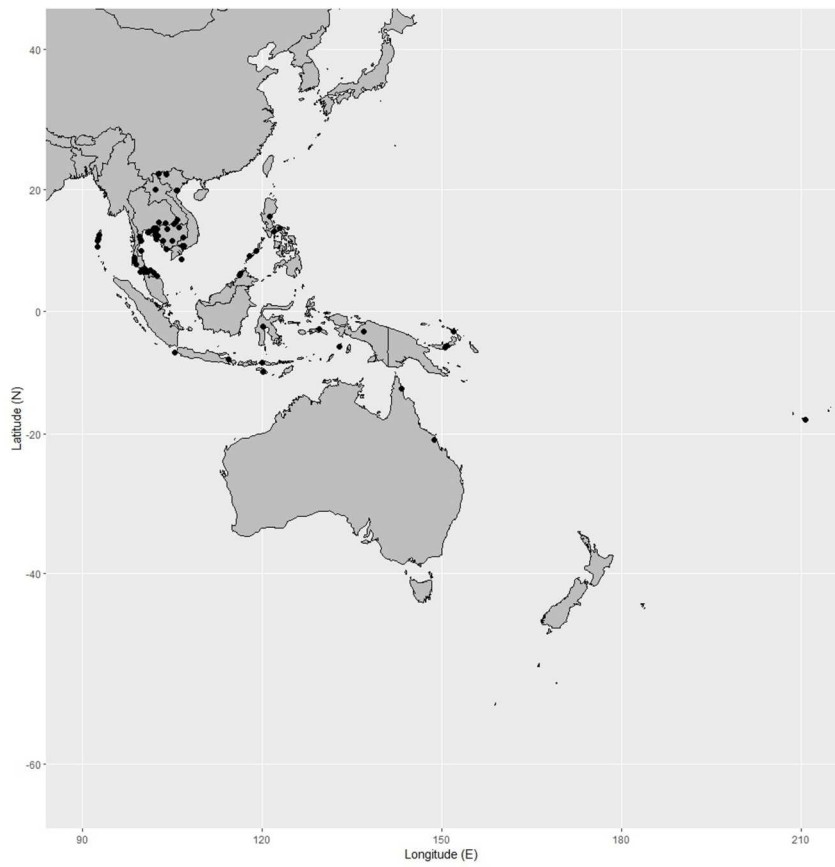

**Figure 6** **Distribution of *Garcinia dulcis*, known from India (including Andaman Islands), Indo-China, Thailand, the Malesian region to Australia (Queensland) and French Polynesia.** Map: Pichet Chanton & Chatchai Ngernsaengsaruay.

*Maheshwari, 1964*; *Allen, 1965*; *Burkill et al., 1966*; *Verheij & Coronel, 1992*). In Java and Singapore, pounded seeds are applied to cure swellings. In Java, the bark is used to dye mats (*Sastri, 1956*; *Maheshwari, 1964*; *Burkill et al., 1966*; *Verheij & Coronel, 1992*).

**Lectotypification.** *Xanthochymus dulcis* was named by *Roxburgh (1820*: 66–67. t. 270), who reported that the species is a native of the Molucca Islands and from thence introduced to the Botanic Garden at Calcutta. This species was transferred to the genus *Garcinia* by *Kurz (1874*: 88). Roxburgh's protologue of *X. dulcis* implies that he based his description on a tree living in the Botanical Garden at Calcutta before he departed India in 1813. No herbarium specimens were cited. Following advice in *Forman (1997)*, in addition to relevant specimens Roxburgh's illustration (Plate 270) in "*Plants of the Coast of Coromandel*" should be considered as a lectotype.

*Garcinia vilersiana* was named by *Pierre (1882*: 348; 1883: t. 71), who cited four gatherings, *Pierre 128*, *Pierre 773* and *Pierre 3641* from Vietnam and *Pierre 3642* from Thailand and Vietnam but he did not mention the herbaria in which they were present, and following Art. 9.6 of the ICN (*Turland et al., 2018*), they constitute syntypes. We could locate the specimens *Pierre 128* (in montibus Dinh ad Baria Austro-Cochinchinae)

at C [without barcode], K [K000677686], L [L0700329, L0700330, L0700331] and P [P04701129, P04899814] collected on October 1866; at BM [BM000611613] collected on November 1866; at P [P04701128, P04701131, P04701136] collected on March 1867, *Pierre 773* (in planitie ad Thu Duc Austro-Cochinchinae) at K [K000742482] and P [P04450818, P04451355, P04458339, P06137125] collected on February 1870; at K [K000677684], L [U1199409] and P [P00903311, P00903312, P04701110, P04898863] collected on February 1871, *Pierre 3641* at P [P04701126, P04701127] (in montibus Dinh ad Baria Austro-Cochinchinae, collected in 1866); at L [U1199410], P [P04701122] (in montibus Dinh ad Baria Austro-Cochinchinae, collected in 1867); at BM [BM000611612], K [K000742481, K000677685] and P [P04899815] (in montibus Dinh ad Baria Austro-Cochinchinae, collected on March 1868); at P [P04701123, P04701124] (in montibus Lu in prov. Bien Hoa Austro-Cochinchinae, collected on March 1877) and *Pierre 3642* from Thailand at P [P04701111, P04701114, P04701115, ad flumen Mekong in regno Siamico, collected on June 1868]; and from Vietnam at P [P04701120, P04701121, in insula Phu Quoc, collected on July 1877]. Hence, the specimen *Pierre 3642* at P [P04701111] is in the best condition and clearly shows the diagnostic characters for the species and is selected here as the lectotype, following Art. 9.3 and 9.12 of the ICN (*Turland et al., 2018*).

*Garcinia cambodgiensis* was named by *Vesque (1889*: 14 t. 92; 1893: 316), who cited the material *Pierre 775* from Cambodia (ad montes Kuang Repoeu, in prov. Tpong) but did not mention the herbaria in which they were present, and following Art. 9.6 of the ICN (*Turland et al., 2018*), they constitute syntypes. We located the specimen *Pierre 775* at K [K000677687] and P [P05062528, P05062544, P05062564], therefore the P [P05062528] material is in the best condition and clearly shows the diagnostic characters for the species and is selected here as the lectotype, following Art. 9.3 and 9.12 of the ICN (*Turland et al., 2018*).

**Additional specimens examined. Thailand. Eastern**: Buri Ram [Nang Rong District, March 1922 [as *Garcinia* sp.], *Luang Anuwat Wanarak 11* (BK, K)], Surin [Cambodia Boundary, Sangkha District, 15 January 1924 [as *G. vilersiana*], *Kerr 8295* (BM, E [E00839801], K, P [P04701109])], Ubon Ratchathani [Huai Don, Buntharik District, 12 May 2005 [as *G. vilersiana*], *Puudjaa 1410* (BKF); Phu Chong Na Yoi National Park, Na Chaluai District, August 2007, *Ngernsaengsaruay G45-10082007* (BKF, dry and spirit materials)]; **South-Western**: Prachuap Khiri Khan [Kui Buri National Park, Kui Buri District, 23 January 2004 [as *G. griffithii*, *G. xanthochymus*], *Middleton et al. 2426* (BKF, E [E00348153]); Pran Buri District, 14 April 2012 [as *G. vilersiana*], *Phengklai 16300* (BKF)]; **Central**: Ang Thong [Wat Klang, Mueang District, 3 April 1972 [as *Garcinia* sp., *G. cf. xanthochymus*], cultivated, *Maxwell 72-180* (AAU, BK); ibid., 7 March 1976 [as *Garcinia* sp., *G. xanthochymus*, *G. vilersiana*], cultivated, *Maxwell 76-135* (AAU, BK, L [L2409257, L2409258])], Saraburi [Phu Khae, 2 April 1948 [as *G. vilersiana*], cultivated, *Chamrueangsri 14* (BKF)], Pathum Thani [locality not specified, 23 July 1979 [as *G. vilersiana*], cultivated, *Muangnoicharoen s.n.* (BKF68140); Suan Pathum Palace, 1 February 2010 [as *G. xanthochymus*], cultivated, *Krajangvuthi s.n.* (BKF163988)], Nonthaburi [Bang Yai District, 1 December 1987 [as *G. vilersiana*], cultivated, *Paisooksantivatana Y1-12-87(1)* (BK)], Bangkok [locality not specified, 13 June 1920, cultivated, *Kerr s.n.* (BM); locality not

specified, 29 January 1922 [as *G. vilersiana*], cultivated, *Marcan 662* (BM, SING); locality not specified, 21 January 1923 [as *G. vilersiana*], cultivated, *Collins 1144* (BM); locality not specified, 4 March 1923 [as *G. vilersiana*], cultivated, *Kerr s.n.* (BM); Rat Burana, 1 December 1987 [as *G. vilersiana*], cultivated, *Paisooksantivatana Y1-12-87(2)* (BK); Faculty of Forestry, Kasetsart University, Chatuchak District, 15 March 1994 [as *G. vilersiana*], cultivated, *Santi 33* (BK); ibid., 4 June 2023, cultivated, *Ngernsaengsaruay & Chanton G50-04062023* (BK, BKF, dry and spirit materials); locality not specified, 27 January 1992 [as *G. vilersiana*], cultivated, *Niyomdham s.n.* (BKF126881); Nong Chok District, 2 July 2007, cultivated, *Ngernsaengsaruay G44-02072007* (BKF, spirit material); Bangkok Herbarium, Department of Agriculture, Chatuchak District, 12 June 2023, cultivated, *Ngernsaengsaruay & Chanton G51-12062023* (BK, BKF)], Samut Prakan [Song Khanong Subdistrict, Phra Pradaeng District, 25 March 2012, cultivated, *Ngernsaengsaruay G47-25032012* (BK, BKF, dry and spirit materials), Samut Songkhram [Kradangnga Subdistrict, Bang Khonti District, 1 December 1987 [as *G. vilersiana*], cultivated, *Paisooksantivatana Y1-12-87(4)* (BK); Amphawa District, 16 December 2003, cultivated, *Bamrungsri 01* (PSU)]; **South-Eastern**: Sa Kaeo [Ban Dong Yang, Aranyaprathet District, 22 March 1962 [as *G. vilersiana*, *G.* cf. *xanthochymus*], *Chantanamuck 95* (BK); Watthana Nakhon District, 5 March 1977 [as *G. vilersiana*], *Unknown s.n.* (BKF110455); Khao Ang Rue Nai Wildlife Sanctuary, 18 January 1997 [as *G. vilersiana*], *Santisuk et al. s.n.* (BKF201290)], Prachin Buri [Kabin Buri District, 22 December 1924 [as *Garcinia* sp., *G. cambodgiensis*, *G. xanthochymus*], *Kerr 9738* (AAU, BK, BM, E [E00839759], K)], Chachoengsao [Khao Takrup, 6 November 1993 [as *Garcinia* sp., *G. xanthochymus*], *Larsen et al. 44255* (AAU, K)], Chon Buri [Si Racha Forest, 1916 [as *G. cambodgiensis*], *Collins 609* (K); Nong Nam Khiao, Si Racha Forest, 1 December 1927 [as *Garcinia* sp., *G. cambodgiensis*, *G.* cf. *xanthochymus*], *Collins 1856* (BK, BM, K); Phan Sadet Nok, Si Racha District, 13 January 1946 [as *Garcinia* sp.], *Din 155* (BKF); Khao Khiao, Si Racha District, 20 March 1970 [as *Garcinia* sp., *G. cambodgiensis*, *G. xanthochymus*], *van Beusekom & Santisuk 3266* (AAU, BKF, C, E [E00772041], K, L [L2409500, L2409501], P [P05062012]); ibid., 5 January 1975 [as *G. cambodgiensis*, *G. xanthochymus*], *Maxwell 75-2* (AAU, BK, L [L2408878]); ibid., 18 May 1975 [as *Garcinia* sp., *G. xanthochymus*], *Maxwell 75-529* (AAU, BK, [L (L2409260)]); Khao Khiao Open Zoo, 9 July 2000, *Phengklai et al. 12659* (BKF)], Chanthaburi [Makham District, 7 August 1933 [as *Garcinia* sp.], *Winit 418* (BKF); Khao Sa Bap, Laem Sing District, 11 November 1945 [as *Garcinia* sp.], *Wit 117* (BKF); ibid., 20 June 1946 [as *Garcinia* sp.], *Wit 279* (BKF); Laem Sing District, 22 December 1961 [as *G. cambodgiensis*, *G vilersiana*], *Nicolson 1645* (K, L [L409326]); Khao Soi Dao, 18 December 1974 [as *Garcinia* sp., *G. cambodgiensis*, *G. xanthochymus*], *Geesink, Hiepko & Phengklai 7904* (BKF, C, K, P [P05062000]); Khao Soi Dao, Pong Nam Ron District, 5 May 1975 [as *Garcinia* sp., *G. cambodgiensis*, *G. xanthochymus*], *Maxwell 75-487* (AAU, BK, L [L2408876, L2408877]); Khao Soi Dao Wildlife Sanctuary, Khao Soi Dao District, 14 March 1995 [as *G.* cf. *vilersiana*], *Santisuk s.n.* (BKF100190); Khao Soi Dao Wildlife Sanctuary, 11 January 1999 [as *G.* cf. *vilersiana*], *Wongprasert et al. s.n.* (BKF124721); Khao Soi Dao Wildlife Sanctuary, 1 March 2007 [as *G. nervosa*, *G. xanthochymus*], *Mantharanon s.n.* (BKF146453); Khao Soi Dao Wildlife Sanctuary, 15 March 2008, *Ngernsaengsaruay G46-15032008* (BKF, spirit material); Khao Soi Dao Wildlife

Sanctuary, 17 May 2013 [as *G. vilersiana*], *Tagane et al. T1553* (BKF); ibid., 17 May 2013 [as *G. vilersiana*], *Tagane et al. T1596* (BKF); Khao Soi Dao Wildlife Sanctuary, 8 March 2014 [as *Garcinia* sp.], *Tagane et al. T2688* (BKF); Khao Phra Bat, Khao Khitchakut National Park, Makham District, 29 November 1979 [as *Garcinia* sp., *G. xanthochymus*], *Shimizu et al. T-23989* (BKF, L [L2409479]); Thung Phen Forest Protection Unit, Khao Khitchakut National Park, 27 December 1993 [as *G. vilersiana*], *Niyomdham 3485* (BKF); Khlong Saba, Khao Sip Ha Chan, Kaeng Hang Maeo, 11 February 2007 [as *G. xanthochymus*], *Watthana 2221* (QBG)], Trat [Bo Rai, 28 November 1924 [as *G. cambodgiensis*], *Kerr 9461* (BM, K); Dan Chumphon, 21 December 1929 [as *Garcinia* sp., *G. cambodgiensis*, *G. xanthochymus*], *Kerr 17657* (AAU, BK, BM, K); Ban Salak Phet, Ko Chang, 7 March 2003 [as *G. mangostana*], cultivated, *Phengklai et al. 14528* (BKF)]; **Peninsular**: Surat Thani [Ko Tao, 18 April 1927 [as *Garcinia* sp., *G. vilersiana*], *Kerr 12797* (BM, K); locality not specified, 1 December 1987 [as *G. vilersiana*], *Paisooksantivatana Y1-12-87(5)* (BK); Thung Khai Han, Khao Sok National Park, 14 June 1994 [as *G. vilersiana*], *Niyomdham & Puudjaa 3786* (BKF); Giant Bamboo nature trail, Khlong Phanom National Park, Phanom District, 20 June 2004 [as *Garcinia* sp., *G. vilersiana*], *Gardner & Sidisunthorn ST0804* (K); ibid., 20 March 2005 [as *Garcinia* sp., *G. vilersiana*], Gardner, *Sidisunthorn & Tippayasri ST1707* (BKF, K)], Krabi [locality not specified, 4 April 1930 [as *G. vilersiana*, *G.* cf. *xanthochymus*], *Kerr 18845* (BK, BM, C, K, L [L2409325]); Ko Pu, 14 April 1930 [as *G. vilersiana*, *G.* cf. *xanthochymus*], *Kerr 18972* (BK, BM, K, L [L2409324]); Ao Luek, 23 July 1972 [as *Garcinia* sp., *G. vilersiana*], *Larsen et al. 31282* (AAU, BKF, K); Than Bok Korani National Park, 9 May 1973 [as *Garcinia* sp., *G. vilersiana*], *Geesink & Santisuk 5303* (AAU, BKF, C, E [E00839800]); Than Bok Korani National Park, Ao Luek District, 7 March 2022, *Ngernsaengsaruay, Meeprom & Boonthasak G49-07032022* (BK, BKF, dry and spirit materials), Trang [Khao Chong Botanical Garden, Chong Subdistrict, Na Yong District, 17 March 2018, cultivated, *Ngernsaengsaruay, Wessapak, Meeprom & Boonthasak G48-17032018* (BK, BKF, dry and spirit materials)], Stun [Ban Ton, 15 March 1928 [as *Garcinia* sp.], *Kerr 14598* (BM, K); Tarutao National Park, trail from Malacca Creek to Ao Talo Wao, 24 March 1980 [as *Garcinia* sp., *G. xanthochymus*], *Congdon 501* (AAU, PSU); near Nam Ra Village, Thung Nui Subdistrict, Khuan Kalong District, 6 January 1985, *Maxwell 85-43* (BKF, PSU); Thale Ban National Park, Khuan Don District, 25 May 2004 [as *Garcinia* sp.], *Gardner & Setsin ST0604* (K)], Songkhla [Namtok Boriphat Forest Park, Rattaphum District, 24 October 1985, *Maxwell 85-997* (AAU, BKF, L [L2402922, L2402923], PSU); Rattaphum District, 15 May 2002, *Upho UBON981* (QBG); Hat Yai District, 6 April 2004, *Deachathai 02* (PSU)]; Hat Yai District, 6 April 2004, *Mahabusarakam 02* (PSU); Khao Nam Khang National Park, Na Thawi District, 6 October 2004, *Tippayasri & Sidisunthorn ST1064* (K); Hat Yai, Khuan Mot Daeng, PSU Campus, 30 May 2005 [as *G. nervosa* ], *Panwiriyarat 01* (PSU)], Pattani [Mueang Pattani District, October 1937, *Unknown 32* (BKF221); Tugong Village, Nong Chik District, 14 January 1985 [as *G. nervosa* var. *pubescens*], *Saree 1* (P [P04701559], PSU), Narathiwat [Ao Manao, 8 August 1999, *Niyomdham & Puudjaa 5753* (BKF); Ao Manao, Khao Tanyong National Park, 27 March 2002, *Ngernsaengsaruay G43-27032002* (BKF, spirit material); Chon Thara Singhe, Tak Bai District, 17 March 2002, cultivated, *Ngernsaengsaruay G42-17032002* (BKF, spirit material)].

**India**. Bengal, s.d., [as *G. xanthochymus*], *Unknown 257* (E [E00772050]); locality not specified, s.d., [as *G. xanthochymus*], *Stewart's Collection 311* (E [E00839674]); South Andaman, 1867 [as *X. pictorius*], *Kurz 241* (P [P04701158]); South Andaman, 1867 [as *X. pictorius*], *Kurz s.n.* (G [G00458419]); Flora of the Andamans, 1884 [as *G. andamanica*], *King's Collector s.n.* (G [G00458474], L [L2408492], P [P05062493]); South Andaman, 1890 [as *G. andamanica*], *King's Collector s.n.* (E [E00438019], L [L2408491]); South Andaman, 7 April 1894 [as *G. andamanica*], *King's Collector s.n.* (P [P05062497]); Little Andaman, 10 January 1976 [as *G. andamanica*], *Bhargava 3382* (L [L2408490]); Middle Andaman, 4 November 1977 [as *G. xanthochymus*], *Bhargawa & Nooteboom 6343* (L [L2409259]).

**Vietnam**. Unreadable, April 1863 [as *G. xanthochymus*], *Pierre 3385* (P [P04701165]); Phu Quoc in Gallicae Austro-Cochinchina, May 1865 [as *G. xanthochymus* ], *Pierre 3385* (L [L2409238]); in insula Condor, October 1876 [as *G. vilersiana*], *Harmand 920* (P [P04701105, P04701106, P04701107]); Tonkin, December 1889 [as *G. cambodgiensis*], *Balansa 4345* (P [P05062548, P05062562]); Bieu Hieu, Thanh Hoa, 10 June 1892 [as *G. cambodgiensis*], *l'abbé Bon 5398* (P [P05062536]); Bien Hoa, 28 February 1914 [as *G. vilersiana*], *Fleury 31300* (P [P04701116]); Bien Hoa, 28 March 1914 [as *G. vilersiana*], *Fleury 32046* (P [P04701119]); Phu Tho, 30 May 1918 [as *G. cambodgiensis*], *Fleury 37545* (P [P05062534]); Annam, Dac Kiet, prov. de Thanh Hoa, 10 September 1920 [as *G. cambodgiensis*], *Poilane 1815* (K, P [P05062532]); Tonkin, Lao Cai, 4 January 1931 [as *G. cambodgiensis*], *Poilane 18764* (P [P05062541]); Lai Chau et Muong, 19 April 1936 [as *G. cambodgiensis*], *Poilane 25747* (P [P05062527]); Tonkin, Lai Chau, 12 January 1938 [as *G. cambodgiensis*], *Poilane 27115* (K, P [P05062542]); Plantes du Tonkin occidental, s.d. [as *G. cambodgiensis*], *l'abbé Bon 5362* (P [P05062533, P05062566]).

**Laos**. Luang Phrabang, expedition du Mekong, 1866–1868 [as *G. cambodgiensis*], *Thorel 9189* (P [P05062546]); Haut-Mékong, 19 May 1936 [as *G. cambodgiensis*], *Poilane 26214* (K, P [P05062535]); Env. de Pakse, Sedone, 18 November 1965 [as *G.* aff. *vilersiana*], *Vidal 4477* (P [P04701130]).

**Cambodia**. Expedition du Mekong, 1866–1868 [as *G. cambodgiensis*], *Thorel 3183* (P [P05062529, P05062530, P05062543]); expedition du Mekong, 1866–1868 [as *G. vilersiana*], *Thorel s.n.* (P [P04701133]); locality not specified, November 1881 [as *G. cambodgiensis*], *Pierre 4171* (P [P04700185]); locality not specified, 1896 [as *G. vilersiana*], *Hahn s.n.* (P [04701134]); Khsach Kandal District, Kandal Province, 7 March 1914 [as *G. vilersiana*], *Unknown 31838* (P [P04701132]); locality not specified, 24 June 1929 [as *G. vilersiana*], *Bejeaud 532* (P [P04701135]); Krapoeu, 26 June 1930 [as *G. cambodgiensis*], *Poilane 17711* (P [P05062545]); Ko Kong, 28 February 2000 [as *G. cambodgiensis*], *Meng 94* (K); Keo Seima District, Mondulkiri Province, 21 May 2001 [as *G. vilersiana*], *Eanghourt & Phirun 851* (K); Siem Reap, Phnom Kulen, 2 July 2006 [as *G. vilersiana*], *Long, Cheng & Leti CL265* (P [P00626101]); Ko Kong, Thma Baing, Veal Kandevech, 19 December 2008 [as *G. cambodgiensis*], *Newman et al. 2069* (E [E00319084]); Stung Treng, Thala Borivat District, 26 March 2012 [as *G. cambodgiensis*], *Maxwell 12-52* (L [L4311881, L4311882]).

**Cochinchine**. Country and locality not specified, 1862–1866 [as *G. vilersiana*], *Thorel 9189* (P [P04701125]); Country and locality not specified, 1862–1866 [as *G. vilersiana*], *Thorel s.n.* (P [P04701108]).

**Indonesia**. Java, Ujung Kulon National Park, 28 November 1960, *Kostermans 241* (L [L2403308]); Java, 30 April 1974, *Wiriadinata 80* (AAU); Lesser Sunda Islands, Sumbawa, 4 October 1982, *Danimihardja SD2217* (L [L2403235]); Lesser Sunda Islands, East Sumba, Ngallu, 5 September 1994, *McDonald & Sunaryo 4402* (E [E00037289]); Sulawesi, 4 July 1976, *Meijer 10817* (L [L2403229]); Moluccas, Ceram, 13 December 1990, *Burley, Tukirin & Ismail 4393* (E [E00160942]).

**Borneo**. Malaysia: Sabah, Jesselton District, 6 May 1963, *Hashim 33874* (L [L2403302]); Sabah, Kampung Payas, Pitas District, 11 May 1987, *Amin et al. SAN121250* (E [E00160943]).

**Philippines**. Palawan, 12 June 1994, *Soejarto et al. 8258* (L [L3813146]); Luzon, 22 February 1991, *Loher 74* (US [US351445]); Malapackun, 15 April 1984, *Ridsdale* SMHI442 (L [L2403249]); Camarines Sur, Lupi, s.d., *Vidal 639* (AAU); Angat, Bulacan, s.d., *Vidal 640* (AAU); Buenavista, Marinduque, s.d., *Vidal 1155* (AAU).

**New Guinea**. Mt. Klangal, Kandrain Subdistrict, West New Britain District, 16 May 1973, *Croft & Katik NGF15591* (US); New Ireland, Ugana, Songmum, West Coast, January 1938, *Peekel 30* (L2403148); Kombi Subdistrict, West New Britain District, 28 May 1973, *Isles & Katik NGF32248* (L2403218).

**Australia**. Queensland, Cook District, 11 October 1962, *Smith 11719* (L2403506); Queensland, Cook District, 12 October 1962, *Smith 11854* (L2403507).

**French Polynesia**. Papeari, Tahiti Islands, 29 November 1963, *Maclet 18* (US).

**2. Garcinia nervosa** (Miq.) Miq., Ann. Mus. Bot. Lugduno-Batavi 1: 208. 1864; Pierre, Fl. Forest. Cochinch. 1(5): 5. 1883; King, J. Asiat. Soc. Bengal, Pt. 2, Nat. Hist. 59(2): 169. 1890; Vesque in A. DC. & C. DC., Monogr. Phan. 8: 327. 1893; Merr., Philipp. J. Sci., C 10(5): 325. 1915; Ridl., Fl. Malay Penins. 1: 179. 1922; Merr., Enum. Philipp. Fl. Pl. 3: 86. 1923; Corner, Wayside Trees Mal. 1: 318. ed. 2, 1952; Whitmore in Whitmore, Tree Fl. Malaya 2: 217. 1973; H. Keng, Concise Fl. Singapore: 49. 1990; N. P. Singh in B. D. Sharma & Sanjappa, Fl. Ind. 3: 121. 1993; I. M. Turner, Gard. Bull. Singapore 47(1): 262. 1995; S. Gardner, P. Sidisunthorn & Chayam., Forest Trees S. Thailand 1: 358. fig. 548. 2015.—*Stalagmitis nervosa* Miq., Fl. Ned. Ind., Eerste Bijv. 3: 496. 1861. Type: Indonesia, Sumatra, Pariaman (originally "Priaman" on the label), s.d., *Diepenhorst HB647* (lectotype selected here L [U0002403, photo seen]; isolectotype L [U0002404, photo seen]) (Fig. 7).

— *Garcinia andersonii* [as *andersoni*] Hook. f. ex T. Anderson in Hook. f., Fl. Brit. India 1: 270. 1874; Vesque in A. DC. & C. DC., Monogr. Phan. 8: 318. 1893. Type: Peninsular Malaysia, Malacca, s.d., *Maingay 157* (distributed at K in 1871) (lectotype selected here CAL [CAL0000005828, photo seen]; isolectotype K [K000677676!]) (Fig. 8).

— *Garcinia macrophylla* T. Anderson in Hook. f., Fl. Brit. India 1: 270. 1874, nom. inval.

—*Garcinia spectabilis* Pierre, Fl. Forest. Cochinch. 1(5): 3. 1883; Vesque, Epharmosis 2: 15 t. 91. 1889. Type: Borneo, 1865, *Beccari 2966* (lectotype selected here P [P04700284!]; isolectotype K [K000677704!]), **syn. nov.** (Fig. 9).

—*Garcinia nervosa* (Miq.) Miq. var. *pubescens* King, J. Asiat. Soc. Bengal, Pt. 2, Nat. Hist. 59(2): 169. 1890; Vesque in A. DC. & C. DC., Monogr. Phan. 8: 327. 1893; Whitmore in Whitmore, Tree Fl. Malaya 2: 218. 1973; I. M. Turner, Gard. Bull. Singapore 47(1): 262. 1995. Type: Peninsular Malaysia, Perak, Larut, 1882, *Kunstler 3197* (lectotype selected

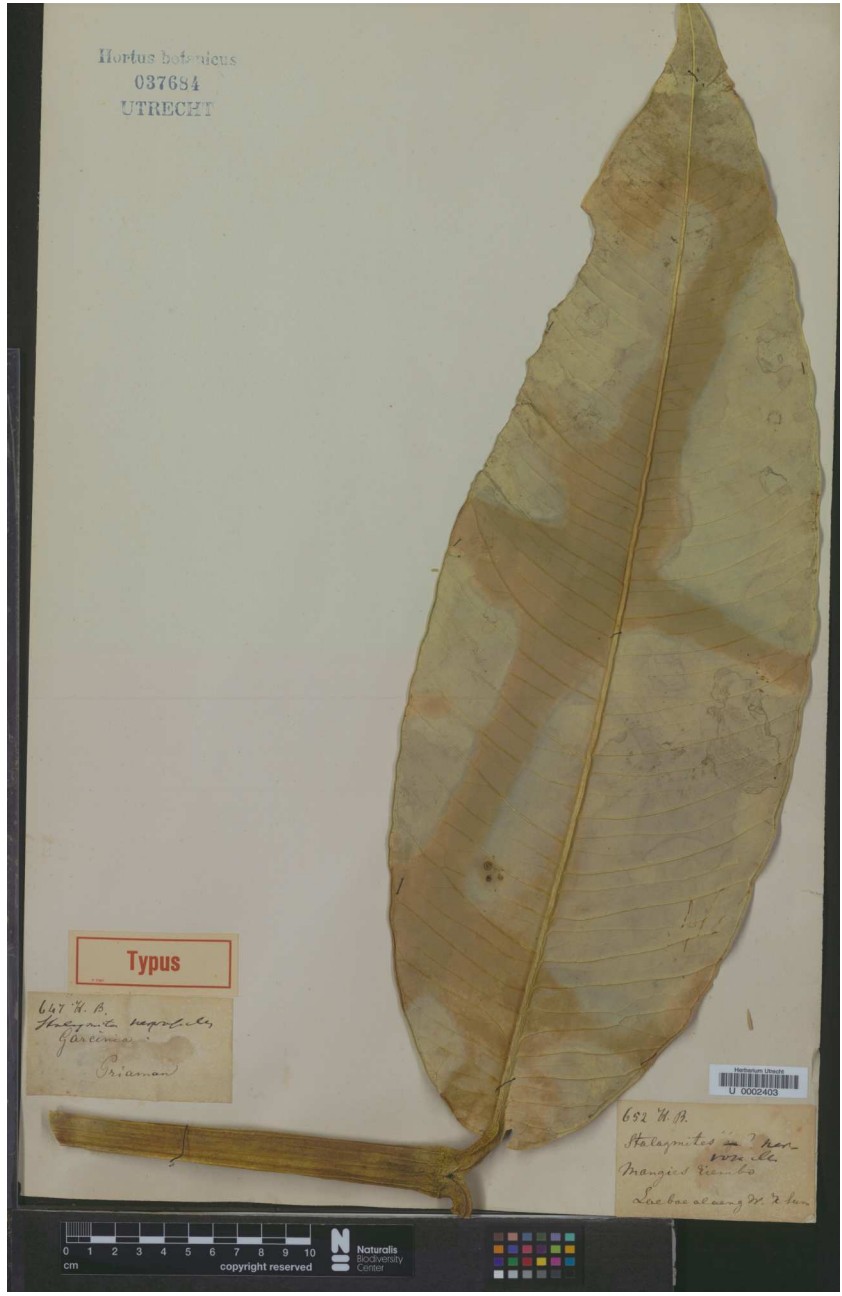

**Figure 7** **Lectotype of *Garcinia nervosa*, *Diepenhorst HB647* (L [U0002403]) from Indonesia, Sumatra, Pariaman (originally "Priaman" on the label).** Photo: https://bioportal.naturalis.nl/en/specimen/U_ _0002403, License: CC0 1.0.

here CAL [CAL0000005834, photo seen]; isolectotypes G [G00458441, photo seen], K [K000677677!], P [P05062500!], SING [SING00636112!, SING00636113!]), **syn. nov.** (Fig. 10).

*Tree* evergreen, 10–35 m tall, 60–160 cm girth; latex white, turning creamish white, very sticky; branches decussate, horizontal; branchlets green, 4-ridged, two of these with

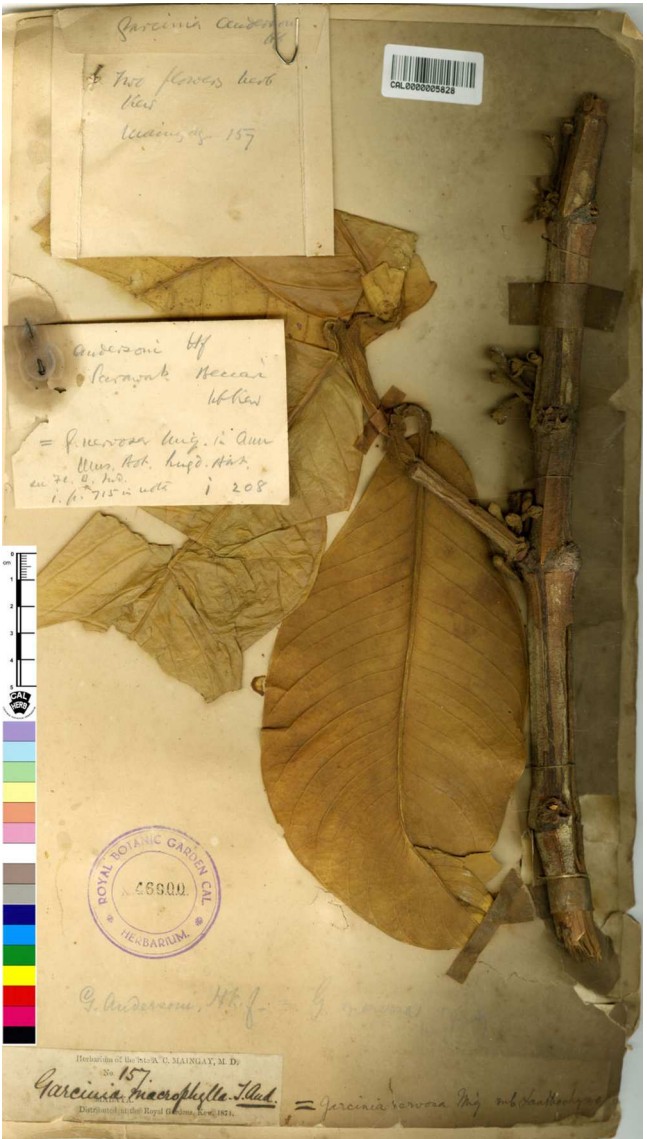

**Figure 8** **Lectotype of *Garcinia andersonii*, *Maingay 157* (CAL [CAL0000005828]) from Peninsular Malaysia, Malacca.** Photo: ©The Director, Botanical Survey of India, Kolkata, https://ivh.bsi.gov.in/phanerogams-Details/en?link=CAL0000005828&column=szBarcode.

a strongly ridged or a narrowly winged, pubescent, slightly pubescent or glabrescent. *Bark* smooth, rough or fine scaly, brown, dark brown or greyish brown; inner bark pale brown. *Terminal bud* concealed between the bases of the uppermost pair of petioles. *Leaves* decussate; lamina narrowly elliptic, narrowly oblong, oblong, lanceolate, lanceolate-ovate or elliptic-oblong, 33.5–80 × 8–27 cm, apex acute, base subcordate, margin repand and slightly revolute, very thickly coriaceous, slightly bullate or bullate, shiny dark green above, paler below, glabrous above, pubescent, slightly pubescent or glabrescent below (of lamina and veins), midrib and secondary veins slightly raised above, strongly ridged

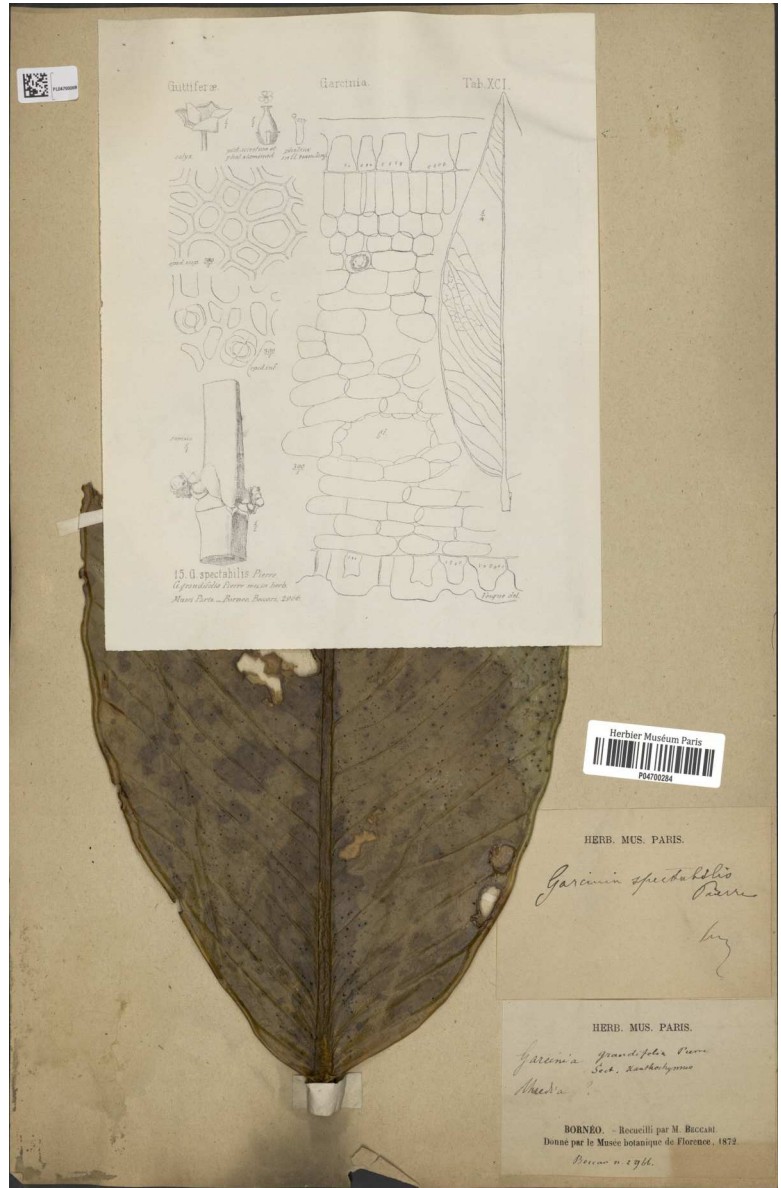

**Figure 9** **Lectotype of *Garcinia spectabilis*, *Beccari 2966* (P [P04700284]) from Borneo.** Photo: Project: RENOBOTA, https://science.mnhn.fr/institution/mnhn/collection/p/item/p04700284 CC BY 4.0, https://creativecommons.org/licenses/by/4.0/.

below, secondary veins 11–24 pairs, curving towards the margin connected in distinct loops and united into an intramarginal vein, sometimes forked, with intersecondary veins, veinlets reticulate, all veins conspicuous on both surfaces, interrupted wavy lines absent; petiole green, the uppermost pair of petioles reddish purple, turning reddish purple tinged with green to green with age, 2.3–7.2 cm long, 0.4–1.1 cm in diam., not grooved, distinctly transversely rugose, indumentum same as in lamina, with a conspicuous basal appendage clasping the branchlet; young leaves shiny pale green; mature leaves turning

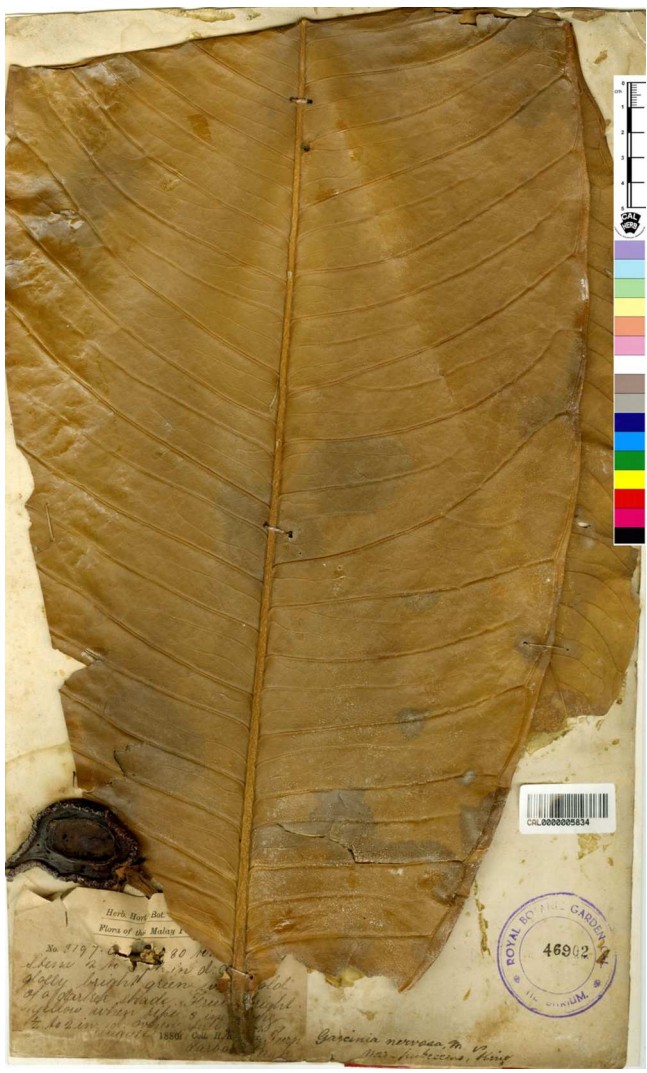

**Figure 10** **Lectotype of *Garcinia nervosa* var. *pubescens, Kunstler 3197*** (CAL [CAL0000005834]) from Peninsular Malaysia, Perak, Larut. Photo: ©The Director, Botanical Survey of India, Kolkata, https://archive.bsi.gov.in/phanerogams-Details/en?link=CAL0000005834&column=szBarcode.

greenish yellow to yellow before falling off; dry leaves yellowish brown. *Inflorescences* on short, leafless lateral branchlets, cymose, in fascicles of 5–15 flowers. *Flowers* unisexual, plants dioecious, 5-merous, petals erect with overlapping edges and corolla forming a bowl-shaped; bracteolate; sepals and petals quincuncial, coriaceous, concave. *Male flowers* 0.8–1.4 cm in diam.; bracteoles triangular, 2–5.2 × 1.8–5 mm, apex acute, pubescent; pedicel green, 1.8–3.2 cm long, widened at the apical part, middle part 3.5–4 mm in diam., apical part 4.5–7 mm in diam., 4-angular, thick, pubescent; sepals 5, reddish green, greenish red, red or green, semi-orbicular or broadly ovate, 3.5–8.5 × 4–9 mm, unequal, apex rounded, pubescent outside; petals 5, whitish pale green, creamish white or pale yellow, suborbicular, obovate or broadly obovate, 0.8–1.4 × 0.5–1 cm, subequal, apex rounded,

margin ciliolate; disk in the center of the flower, intrastaminal, yellow, 5-lobed, fleshy, pitted, lobes positioned between the stamen bundles, antesepalous; stamens numerous, united in 5 bundles, 11–20 in each bundle, antepetalous, 0.8–1 cm × 2–3 mm each bundle, creamish white or pale yellow; filaments 0.5–0.6 mm long; anthers yellow, 2 thecae, 0.3–0.5 × 0.3–0.7 mm; pistillode absent. *Female flowers* 1–1.7 cm in diam.; bracteoles and pedicel same as in male flowers; sepals and petals same as or slightly larger than in male flowers; appendages 5, antesepalous, alternating with staminode bundles, fleshy, pitted; staminodes united in 5 bundles, 4–6 in each bundle, antepetalous, 3–4 mm long, 1–1.5 mm wide each bundle, creamish white or pale yellow; pistil 0.7–1 cm long; ovary reddish green or green, broadly ovoid, subglobose or globose, 3.5–6 × 3.2–5.5 mm, beaked, 1.5–3.5 × 1.2–2 mm (ovary including beaked looks like obpyriform in outline), unlobed, glabrous, 5-locular; stigma red, sessile, radiate, deeply 5-lobed, 3–4 mm in diam., papillate. *Fruits* berries, green or reddish green, turning yellow or yellow blotched with red when ripe, slightly rough, glabrous, not glossy, with sticky yellow latex, then exocarp becoming dark brownish black and sinuously wrinkled when dry, subglobose, globose or broadly ovoid, 5–7.5 × 4.5–7 cm, sometimes oblique, asymmetrical, unlobed, with a short, thick beak; persistent stigma red or greenish red, radiate, deeply 5-lobed, erect; persistent sepals slightly larger than in flowering materials; fruiting stalk green, 3–5.2 cm long, widened at the apical part, middle part 4–6.5 mm in diam., apical part 0.5–1 cm in diam., 4-angular, thick, pubescent. *Seeds* 2–5, sometimes aborted, brown mottled with irregular pale brown lines, semi-ellipsoid, 2.8–3.2 × 1.5–2 cm, rounded at both ends, with a thin white fleshy pulp (Fig. 11).

**Distribution.** India (Andaman and Nicobar Islands), Peninsular Thailand, Peninsular Malaysia [Perak, Terengganu (also called Trengganu), Pahang, Selangor, Malacca (also called Melaka)], Singapore, Indonesia (Sumatra), Borneo (Sabah, Sarawak, Brunei, East Kalimantan, West Kalimantan, South Kalimantan), Philippines (Palawan, Luzon) (Fig. 12).

**Distribution in Thailand.** Peninsular: Nakhon Si Thammarat, Phatthalung, Trang, Satun, Pattani, Yala, Narathiwat (Fig. 12).

**Habitat and Ecology.** It is usually found in tropical lowland evergreen rain forests, sometimes in limestone areas, often near or along streams, 50–250(–600) m alt. In Peninsular Malaysia and Singapore, it occurs in lowland, lowland dipterocarp, hill dipterocarp and freshwater swamp forests, sometimes along streams or near the sea up to elevations of 750 m amsl. (*Corner, 1952*; *Whitmore, 1973*; the specimens from Peninsular Malaysia and Singapore).

**Phenology.** Flowering and fruiting more than once; flowering nearly throughout the year, usually in January to April; fruiting February to August.

**Conservation Status.** *Garcinia nervosa* is widely distributed from Andaman and Nicobar Islands to Philippines (*POWO, 2023*) and is very rare in Singapore (*Keng, 1990*); the specimen *Sinclair 10915*. It has a large EOO of 3,215,117.07 km$^2$ and a relatively large AOO of 292 km$^2$. In Thailand, this species is known to be naturally distributed in the peninsular region, and has an EOO of 25,909.66 km$^2$ and an AOO of 40 km$^2$. There doesn't appear to be an imminent threat to the plants or their habitats. Therefore, we consider the conservation assessment here as LC.

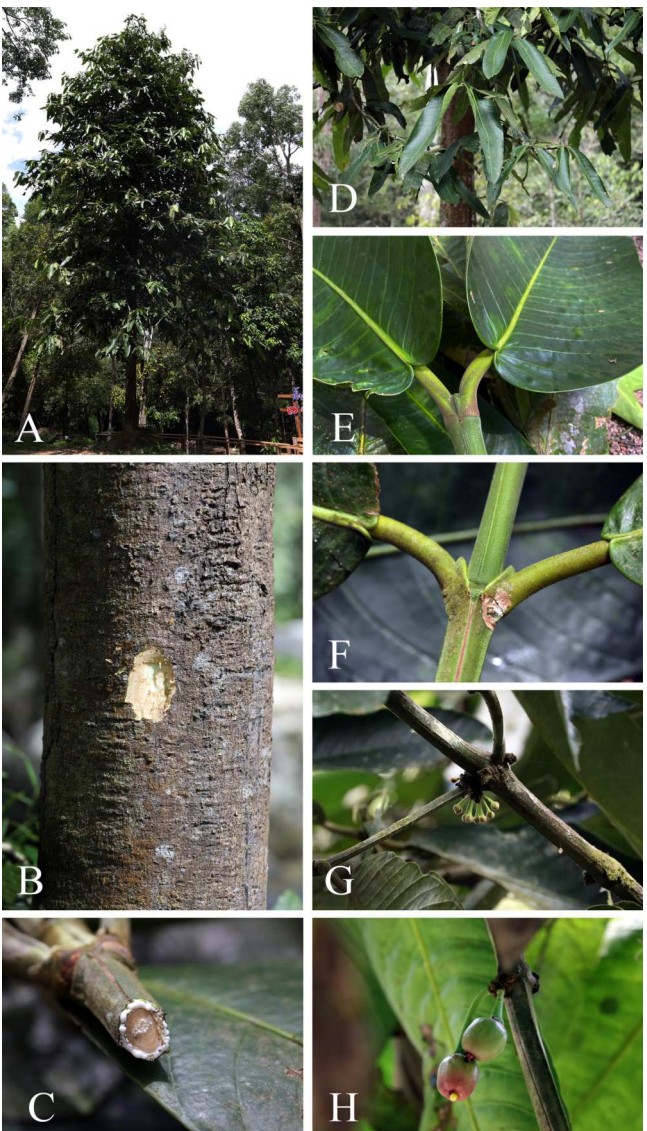

**Figure 11** *Garcinia nervosa.* (A) Habit. (B) Stem, outer bark, inner bark and slashed bark with white latex. (C) Cut branchlet with white latex. (D) Branchlets and leaves. (E) Terminal bud concealed between the bases of the uppermost pair of petioles. (F) Petiole with a conspicuous basal appendage clasping the branchlets. (G) Inflorescences on short, leafless lateral branchlets with young flower buds. (H) Fruiting branchlets with very young fruits. Photos: Chatchai Ngernsaengsaruay.

**Etymology.** The specific epithet of *Garcinia nervosa* is a Latin word, referring to the strongly or prominently nerved (veined) leaves (*Stearn, 1992*; *Gledhill, 2002*). The specific epithet of its new synonym *G. spectabilis* is a Latin word meaning admirable, spectacular or good-looking, refers to the character of the leaves (*Gledhill, 2002*).

**Vernacular Name.** Chamuang nam (ชะมวงน้ำ) (Yala); Phut (พูด) (Nakhon Si Thammarat, Phatthalung, Trang); Maphut pa (มะพูดป่า) (Pattani); Mu-lu (มูลู) (Malay-Pattani); Asam

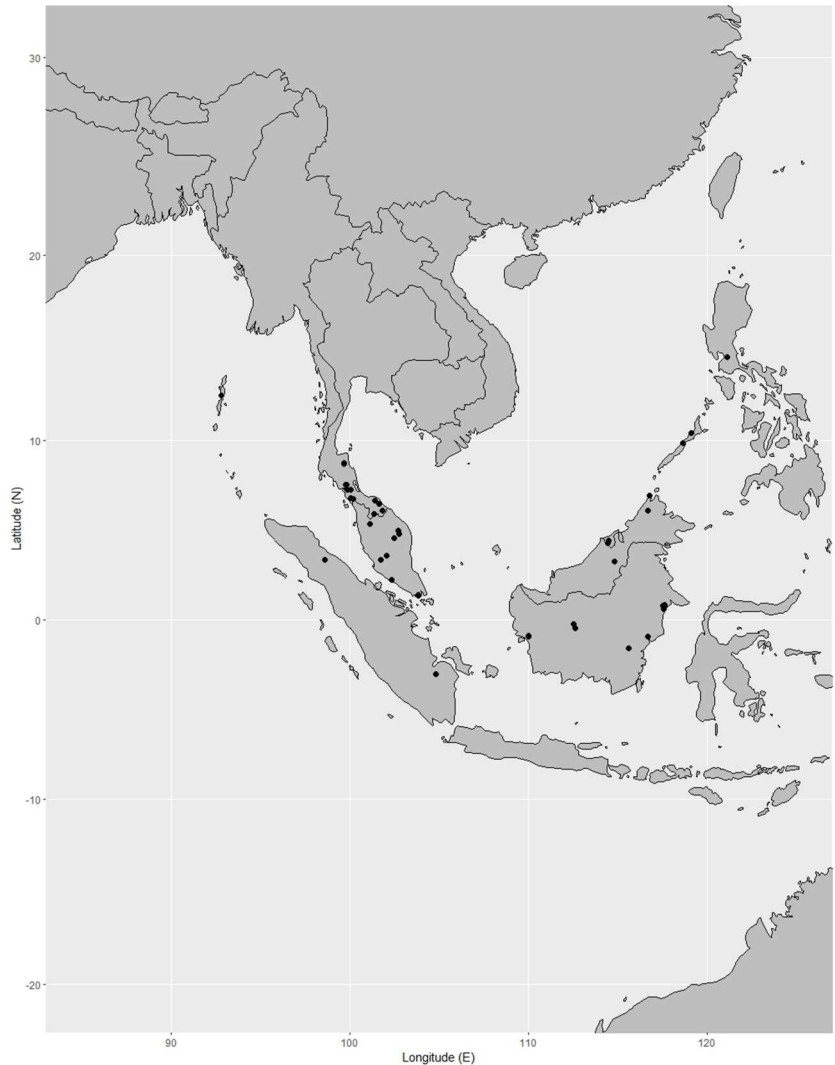

**Figure 12** **Distribution of *Garcinia nervosa*, known from Andaman and Nicobar Islands, Peninsular Thailand to the Malesian region.** Map: Pichet Chanton & Chatchai Ngernsaengsaruay.

garam, Kandis gajah, Pakok lapan taun (Malay); Buradgis, Kabal, Gatatán (Philippines); Pear mangosteen (English).

**Uses.** The fruits (pericarp and fleshy pulp) are edible and have a sour or sweet-sour taste. In Malaysia, the sour pulp is eaten cooked with sugar (*Bircher & Bircher, 2000*). In Andaman and Nicobar Islands, the fruits have been used by the Nicobarese for food and medicinal purposes and the wood is used by the Nicobarese and the Shompen for making canoe paddles (*Waman, Bohra & Mane, 2018*). Leaves are pounded into paste, boiled in coconut oil, and rubbed onto body and joints for pain relief (*National Parks, Flora and Fauna Web, 2023*). The leaves and bark contain high antioxidative and anti-inflammation properties, which have great potential in the development of pharmaceutical and dermatological
products (*Seruji, Khong & Kutoi, 2013*). It is a tree suitable for gardens, parks and roadsides (*National Parks, Flora and Fauna Web, 2023*).

**Lectotypification.** *Stalagmitis nervosa* was named by *Miquel (1861*: 496), who cited two gatherings from Sumatra: D. (Diepenhorst) in "Priaman" and T. (Teysman) in "Lubu-alang" but did not mention the collector number and the herbaria in which they were present. This species was transferred to the genus *Garcinia* by *Miquel (1864*: 208). We only found two sheets of the specimen *Diepenhorst HB647* collected from "Priaman" at L [U0002403, U0002404], and following Art. 9.6 of the ICN (*Turland et al., 2018*), they constitute syntypes. Friedrich Anton Wilhelm Miquel (Miq.) (1811–1871) was a Dutch botanist, a professor of botany and a director of the Amsterdam botanical garden (1846–1859), then a director of the Utrecht Botanical Garden (1859–1871), and from 1862 was a director of the Leiden Rijksherbarium. Miquel's private herbarium, containing many of his types, is the basis of the general herbarium of U. Other Miquel types are, however, in several herbaria because Miquel described many new taxa based on material obtained on loan. Most of these types are at L; others are at G, P, and K. The plants in many herbaria labelled "Ex Herbario Miquel", were not collected by Miquel himself but by various other collectors (*Stafleu & Cowan, 1981*). Therefore, the specimen *Diepenhorst HB647* at L [U0002403] is selected here as the lectotype, following Art. 9.3 and 9.12 of the ICN (*Turland et al., 2018*).

*Garcinia andersonii* (as *G. andersoni*) was named by Joseph Dalton Hooker (Hook. f.) and then described by *Anderson (1874*: 270), who cited the collection *Maingay 157* from Malacca but did not mention the herbaria in which it was present. The authors located two sheets of this specimen at CAL [CAL0000005828] and K [K000677676], and following Art. 9.6 of the ICN (*Turland et al., 2018*), they constitute syntypes. Thomas Anderson (1832–1870) was a Scottish botanist and a superintendent of the Calcutta Botanical Gardens (1860–1868) (*Stafleu & Mennega, 1992*). We therefore selected the CAL specimen as the lectotype, following Art. 9.3 and 9.12 of the ICN (*Turland et al., 2018*).

*Garcinia spectabilis* was named by *Pierre (1883*: 3) based on the specimen *Beccari 2966* from Borneo but he did not mention the herbaria in which it was present. We could locate two sheets of this specimen at K [K000677704] and P [P04700284], and following Art. 9.6 of the ICN (*Turland et al., 2018*), they constitute syntypes. Jean Baptiste Louis Pierre (1833–1905) was a French botanist, a director at the Saigon botanical garden, and explored Cambodia, Cochinchina, and southern Thailand (1865–1877), and returned to France in 1877 (*Stafleu & Cowan, 1983*)). Hence, the P [P04700284] material is selected here as the lectotype, following Art. 9.3 and 9.12 of the ICN (*Turland et al., 2018*).

*Garcinia nervosa* var. *pubescens* was named by *King (1890*: 169), based on the cited specimen *King's Collector 3197* from Larut, Perak but he did not mention the herbaria in which it was present. We traced six sheets of the specimen *Kunstler 3197* at CAL [CAL0000005834], G [G00458441], K [K000677677], P [P05062500] and SING [SING00636112, SING00636113] but only the specimens at K, P and SING were labeled as *King's Collector 3197*. However, all materials are of Hermann H. Kunstler as many of his collections are labelled "*King's Collector*" (*van Steenis-Kruseman & van Steenis, 1950*), and were collected from the same locality. Following Art. 9.6 of the ICN

(*Turland et al., 2018*), they constitute syntypes. Sir George King (1840–1909) was a British botanist, a superintendent of the Calcutta Botanic Gardens (1871–1898), and a director Botanical Survey of India (1891–1898) (*Stafleu & Cowan, 1979*). Therefore, the CAL [CAL0000005834] specimen is selected here as the lectotype, following Art. 9.3 and 9.12 of the ICN (*Turland et al., 2018*).

**Additional specimens examined. Thailand. Peninsular**: Nakhon Si Thammarat [National Park Protection Unit, Krung Ching Waterfall, Khao Luang National Park, Nop Pitam District (originally "Tha Sala District" on the label), 17 March 2005 [as *Garcinia* sp.], *Sidisunthorn & Tippayasri ST1688* (K); ibid., 27 February 2006 [as *Garcinia* sp.], *Gardner & Sidisunthorn ST1688a* (BKF, K)], Phatthalung [Lan Mom Chui Waterfall, Ban Tamot Wildlife Protection Unit, Tamot District, 23 August 1996, *BGO. Staff 7317* (QBG); Tamot Waterfall (also called Lan Mom Chui Waterfall), Tamot Subdistrict, Tamot District, 8 March 2022, *Ngernsaengsaruay & Boonthasak G35-08032022* (BK, BKF); locality not specified, 24 November 2004, *Watcharin 01* (PSU)], Trang [16-ha plot, Khao Chong, February 2001, *Sinbumroong & Davies AS286* (BKF); 24-ha long-term ecological research plot, Khao Chong, Chong Subdistrict, Na Yong District, 16 February 2022, *Ngernsaengsaruay, Wessapak, Meeprom & Boonthasak G34-16022022* (BK, BKF); Ton Te Waterfall, Palian District, February 2001 [as *Garcinia* sp.], *Sinbumroong & Davies AS259* (BKF); Ban Tha Khao, Palian District, 10 July 2004, *Maneenoon & Srimok 698* (PSU); Namtok Phan Forest Park, Palian Subdistrict, Palian District, 17 March 2018, *Ngernsaengsaruay, Wessapak, Meeprom & Boonthasak G33-17032018* (BK, BKF)], Satun [Ya Roi Waterfall, Thaleban National Park, Khuan Don District, 18 March 2004, *Gardner & Sidisunthorn ST0247* (BKF)], Pattani [reported by *Gardner, Sidisunthorn & Chayamarit (2015)*], Yala [Bang Lang National Park, Aiyoeweng Subdistrict, Betong District, 22 May 2022 (Ngernsaengsaruay own observation, with photos); reported by *Gardner, Sidisunthorn & Chayamarit (2015)*], Narathiwat [Chat Warin Waterfall, To Deng Subdistrict, Su-ngai Padi District, 22 April 1995, *Niyomdham 4080* (BKF); Pacho Waterfall, Budo–Su-ngai Padi National Park, Bacho District, 27 December1999 [as *Garcinia* sp.], *Wongprasert 9912-77* (BKF)]; Peninsular, Province not specified [locality not specified, s.d., *Winit 35* (BKF)].

**Peninsular Malaysia**. Perak [Larut, July 1886, *King's Collector 10491* (CAL [CAL0000005829]); Gerik, 3 February 1994, *Penomot & Teo 4334* (P [P04701558])], Terengganu [Tasik Kenyir, Hulu Terengganu, 31 July 2007, *Imin et al. Kep. Forest Research Institute, FRI58501* (L [L3806836])], Pahang [Ulu Keniyam, 3 March 1968, *Shah 1531* (C); Krau Wildlife Reserve, Kuala Lompat, 15 February 1983, *Daveson s.n.* (E [E00839740])], Malacca [Air Kroh, 26 February 1989, *Khairuddin Kep. FRI33158* (L [L3806892]).

**Singapore**. MacRitchie Reservoir, 31 August 1964, *Sinclair 10746* (E [E00839741, E00839742]); Nee Soon, Seletar Forest, 6 September 1966, *Sinclair 10915* (E [E00839738, E00839739]).

**Indonesia**. Sumatra [Sibolangit, 29 March 1908, *Lörzing 5601* (L [L2408483, U1199434]); Eil. Simaloer, 3 August 1918, *Achmad 554* (L [L2408480, U1199435]); ibid., 26 December 1918, *Achmad 813* (L [L2408503, L2408504, U1199431]); ibid., 9 June 1919, *Achmad 1165* (L [L2408488, L2408489, U1199433]); Res. Palembang, 17 April 1920, *Endert 165EIP 857* (L [U1199432])].

**Borneo**. Malaysia: Sabah [Bengkoka Forest Reserve, Kudat, 7 September 1972 [as *G. mangostana*], *Shea & Minjulu SAN75989* (L [L2416697]); TM 1, Ranau, Sabah, 24 February 1990, *Unknown SAN128804* (L [L3806446])]; Indonesia: East Kalimantan [PT. ICTI, road Kenangan to Sepaku, 25 April 1995, *Kessler et al. 896* (L [L2416702]); Kaltim Prima Coal (KPC area), Bengalon, Sebongkok Utara, 8 April 1996, *Arifin et al. AA1699* (L [L2416700, L2416701]); KPC area, Sangatta, 6 April 1997, *Kessler et al. 2386* (L [L2416698])], West Kalimantan [Serawai, 13 October 1995, *Church et al. 2465* (L [L3811160]); Gunung Palung National Park, Ketapang, Kalimantan Barat, 8 October 1997, *Laman et al. 1148* (L [L3813083, L3813084])], South Kalimantan [PT. Aya Yayang, Kabupaten Tabalong, 17 July 2000, *Sidiyasa & Arifin 2160* (L [L2416699])]; Brunei [Labi, Bukit Teraja, Belait, 11 November 1990, *Kirkup 267* (L [L3806485]; Labi, Teraja, Belait, 23 May 1996, *Joffre et al. 17472* (L [L3878579])].

**Philippines**. Palawan [Municipality, San Vicente, 1 April 1990, *Soejarto & Madulid 7215* (L [U1199430]); Taytay, 1913, *Merrill 9387* (P [P04700489, P04700490])].

**3. Garcinia prainiana** King, J. Asiat. Soc. Bengal, Pt. 2, Nat. Hist. 59(2): 171. 1890; Vesque in A. DC. & C. DC., Monogr. Phan. 8: 329. 1893; Ridl., Fl. Malay Penins. 1: 180. 1922; Corner, Wayside Trees Mal. 1: 320. fig. 112. ed. 2. 1952; Corner & Watan., Ill. Guide Trop. Pl.: t. 193. 1969; Whitmore in Whitmore, Tree Fl. Malaya 2: 220. 1973; I. M. Turner, Gard. Bull. Singapore 47(1): 263. 1995. Type: Peninsular Malaysia, Perak, Kuala Dipang (originally "Kwala Dipang" on the label; originally published "Kwala Dynong"), February 1885, *Scortechini 1796* (lectotype selected here CAL [CAL0000005844, photo seen]; isolectotypes K [K000677678!], P [P04701324, photo seen]) (Fig. 13).

*Tree* evergreen, 3–12 m tall, 15–75 cm girth; latex white, sticky; branches decussate, horizontal; branchlets green, terete, glabrous. *Bark* smooth or slightly rough, pale brown, greyish brown or blackish brown; inner bark pale yellow. *Terminal bud* concealed between the bases of the uppermost pair of petioles. *Leaves* decussate; lamina elliptic, oblong or elliptic-oblong, sometimes narrowly oblong, 12.5–27.5 × 5.5–11.5 cm, apex acute or obtuse, base subcordate, often subamplexicaul, margin repand and slightly revolute, coriaceous, bullate or slightly bullate, shiny dark green above, paler below, glabrous on both surfaces, midrib flattened above, raised as a prominent ridge below, secondary veins 9–20 pairs, curving towards the margin connected in distinct loops and united into an intramarginal vein, flattened above, raised below, conspicuous on both surfaces, with intersecondary veins, veinlets reticulate, visible on both surfaces, interrupted long wavy lines of differing lengths, nearly parallel to the midrib, running across the secondary veins to the apex, visible below; petiole green, short, 1.5–6 mm long, 2–5 mm in diam., not grooved, transversely rugose, glabrous, with a basal appendage clasping the branchlet; young leaves shiny pale green. *Inflorescences* terminal, sometimes on short, leafless lateral branchlets, cymose, usually in dense fascicles of several to many flowers. *Flowers* unisexual, plants dioecious, 5-merous, fully opened flowers with spreading petals; bracteolate; sepals and petals quincuncial, coriaceous, glabrous. *Male flowers* 2.5–3.5 cm in diam.; bracteoles pale green, triangular 2.3–4.5 × 1.8–3.7 mm, apex acute, conduplicate with a central keel; pedicel pinkish green, reddish green or greenish red, 3–6 mm long, 2.5–4 mm in diam., widened at the apical part, terete, glabrous; sepals 5, pinkish green, reddish green or greenish

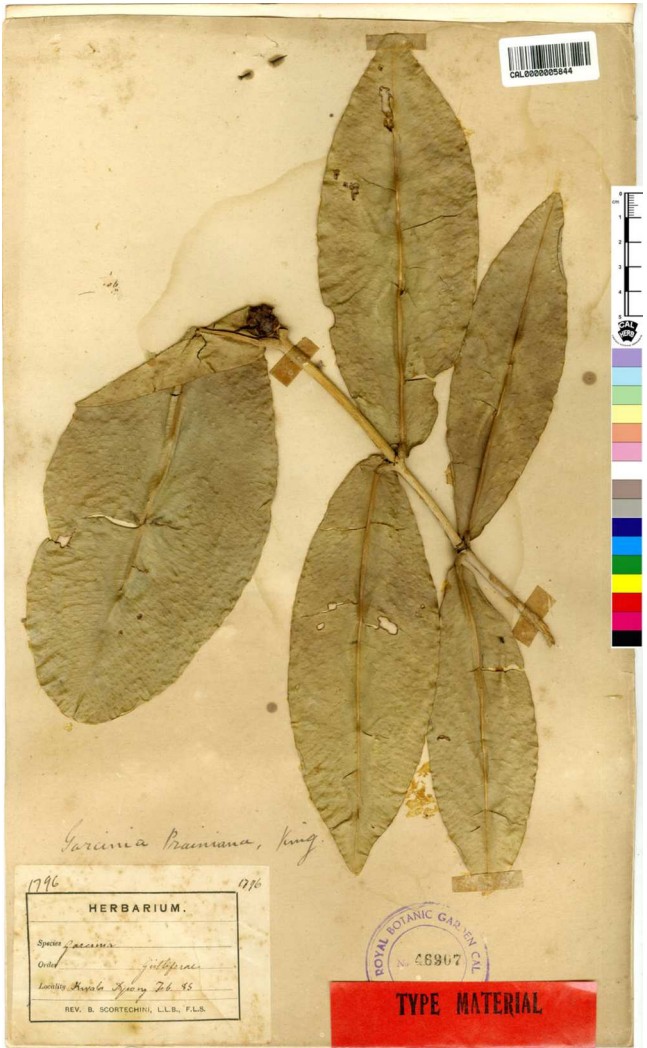

**Figure 13** Lectotype of *Garcinia prainiana*, *Scortechini 1796* (CAL [CAL0000005844]) from Peninsular Malaysia, Perak, Kuala Dipang (originally "Kwala Dipang" on the label; originally published "Kwala Dynong"). Photo: ©The Director, Botanical Survey of India, Kolkata, https://ivh.bsi.gov.in/phanerogams-Details/en?link=CAL0000005844&column=szBarcode.

red, concave, broadly ovate or suborbicular 4.8–8 × 5–7.8 mm, unequal, apex rounded; petals 5, variable in color: pale yellow, yellowish pink, yellowish red, pinkish red, pink or red, broadly obovate or obovate, 0.8–1.4 × 0.6–1.1 cm, subequal, sometimes unequal, apex rounded; a small ring-shaped disk surrounding the base of the pistillode; stamens numerous, united in 5 bundles surrounding the pistillode, antepetalous, 1.7–4.2 mm long, 1.2–4 mm wide each bundle, pale yellow, pink or red; filaments fused throughout their entire length; anthers yellow, 2 thecae, 0.3–0.6 mm long; pistillode mushroom-shaped (fungiform), 5.5–7.5 mm long; sterile stigma pale yellow, pink or red, sessile, convex, indistinctly lobed, 5–6 mm in diam., papillate. *Female flowers* 2.5–4 cm in diam.; bracteoles and pedicel same as in male flowers; sepals and petals same as or slightly larger than in male

flowers; staminodes absent; pistil mushroom-shaped, 6–8.5 mm long; ovary pale green, depressed globose 4–6 × 4.5–6.5 mm, unlobed, glabrous, 5–8-locular; stigma pale yellow, pink or red, sessile, convex, weakly 5–8-lobed or indistinctly lobed, 5–7 mm in diam., papillate. *Fruits* berries, green, turning greenish yellow, bright yellow, orangish yellow and bright orange when ripe, smooth, glabrous, glossy, then exocarp becoming dark brownish black and slightly sinuously wrinkled when dry, depressed globose or depressed subglobose, sometimes globose, 2–3.5 × 2–4.5 cm, sometimes oblique, asymmetrical, unlobed, slightly concave or flattened at the apex, pericarp 3.5–8 mm thick, exocarp thin; persistent stigma dark brown or blackish brown, circular, button-like, 6–9 mm in diam., slightly concave or flattened, weakly 5–8-lobed or indistinctly lobed, papillate; persistent sepals pale green, turning yellowish green and orangish green, larger than in flowering material; fruiting stalk green, 0.4–1 cm long, 3–5 mm in diam., thick. *Seeds* 1–6, often aborted, brown, broadly ellipsoid, ellipsoid or subglobose, 0.9–1.5 × 0.7–1.2 cm, with pale orange fleshy pulp. (Figs. 14 and 15).

**Distribution.** Known only from Peninsular Thailand and Peninsular Malaysia. It is widely distributed in Peninsular Malaysia: Perlis, Kedah, Penang (also called Pulau Pinang), Perak, Kelantan, Terengganu, Pahang, Selangor, Negeri Sembilan (also called Negri Sembilan), Malacca (also called Melaka) and Johor (also spelled Johore) (*Corner, 1952*; *Whitmore, 1973*; *Turner, 1995*; *Azuan & Salma, 2018*). It can be found mainly in Pahang, Perak, and Negeri Sembilan (*Syazwani, 2020*) (Fig. 16).

**Distribution in Thailand.** It is uncommon in Peninsular Thailand: Yala (Than To) and Narathiwat (Waeng and Su-ngai Kolok) (Fig. 16).

**Habitat and Ecology.** It is found in tropical lowland evergreen rain forests, occasionally along streams, 30–200 m alt. It is also cultivated in villages and botanical gardens. In Peninsular Malaysia, it occurs in lowland and hill forests, on hillsides and ridges up to elevations of 1,000 m amsl. It is also cultivated in villages (*Whitmore, 1973*; *Syazwani, 2020*; from the specimen *Whitmore Kep. FRI4018*).

**Phenology.** Flowering and fruiting more than once; flowering nearly throughout the year, usually in February to May; fruiting April to June and September to December. According to *Syazwani (2020)*, in Peninsular Malaysia, the fruits are borne once a year, from July to September.

**Conservation Status.** *Garcinia prainiana* is widely distributed in Peninsular Malaysia and less common in southern Thailand. It has an EOO of 359,696.7 km$^2$ and an AOO of 72 km$^2$ and does not face any threat of extinction. We therefore consider the conservation assessment as LC in agreement with *Kochummen (1998)*.

**Etymology.** The specific epithet of *Garcinia prainiana* refers to Sir David Prain (1857–1944), a British botanist, an herbarium curator of the Royal Botanic Garden, Calcutta (1887–1898), and a director of the Royal Botanic Gardens, Kew (1905–1922) (*Stafleu & Cowan, 1983*)).

**Vernacular Name.** Chupu (ชูปู) (Malay-Narathiwat); Cerapu, Chekau, Chepu, Cherapu, Cherpu, Cherupu, Chupak, Chupu, Kechupu, Kecupu, Menchepu, Menchupu (Malay); Button mangosteen (English).

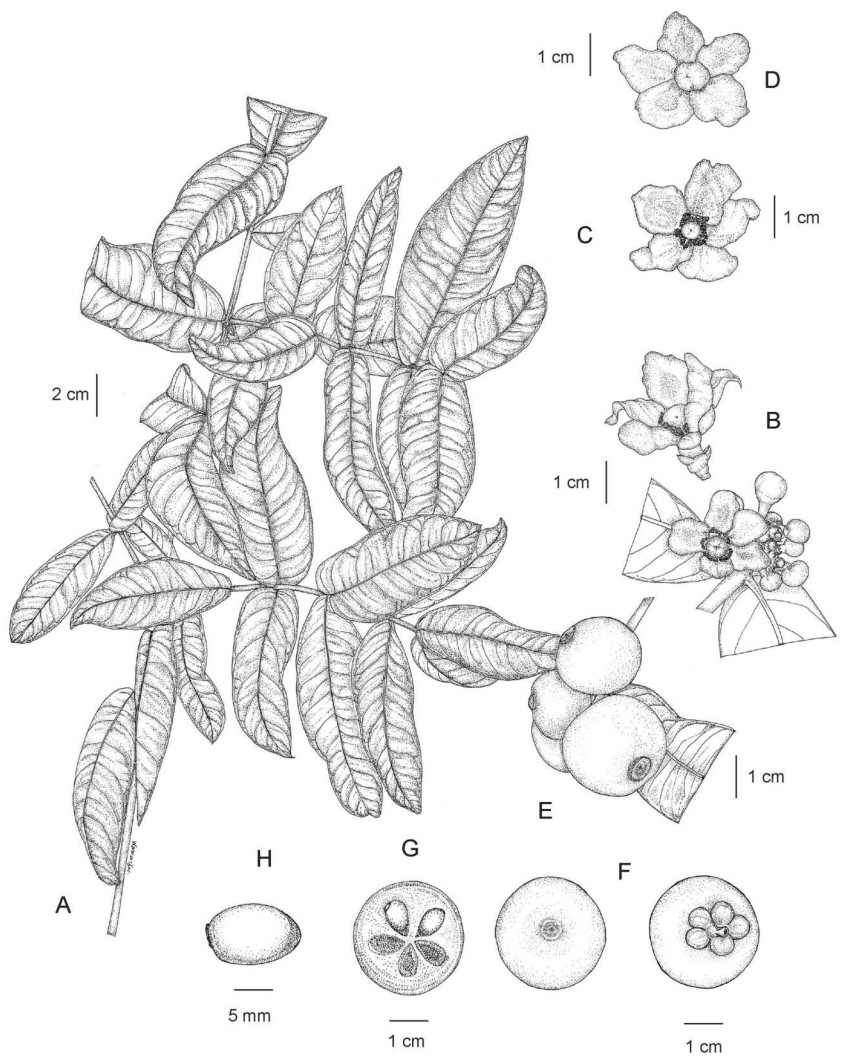

**Figure 14** *Garcinia prainiana.* (A) Branchlets and leaves. (B) Fully opened male flower (side view) and inflorescences with fully opened male flower and male flower buds. (C) Fully opened male flower with spreading petals showing stamens united in 5 bundles and a pistillode (top view). (D) Fully opened female flower with spreading petals showing a pistil (top view). (E) Fruiting branchlet. (F) Fruit with persistent stigma (left) and fruit with persistent sepals (right). (G) Transverse section of fruit with seeds. (H) Seed. Photo: Drawn by Wanwisa Bhuchaisri.

**Uses.** *Garcinia prainiana* is locally cultivated for its fruits in southern Thailand. The fruits (pericarp and fleshy pulp) are edible and have a sour or sweet-sour taste. It is also grown in some botanical gardens as an ornamental plant to provide botanical education. In Peninsular Malaysia, it is commonly cultivated in village gardens. The ripe fruits (fleshy pulp) are edible and are sometimes used fresh in beverages (*Allen, 1965*; *Burkill et al., 1966*). The pulp of fruits of has high antioxidant content of about 91.9% and vitamin C content of about 27.3 mg per 100 g fresh weight (*Azuan & Salma, 2018*). In a traditional Malay recipe, the raw fruits are described as being cooked with dried fish (*Zawiah & Othaman, 2012*). The wood is used for house building (*Allen, 1965*; *Burkill et al., 1966*). It is an excellent

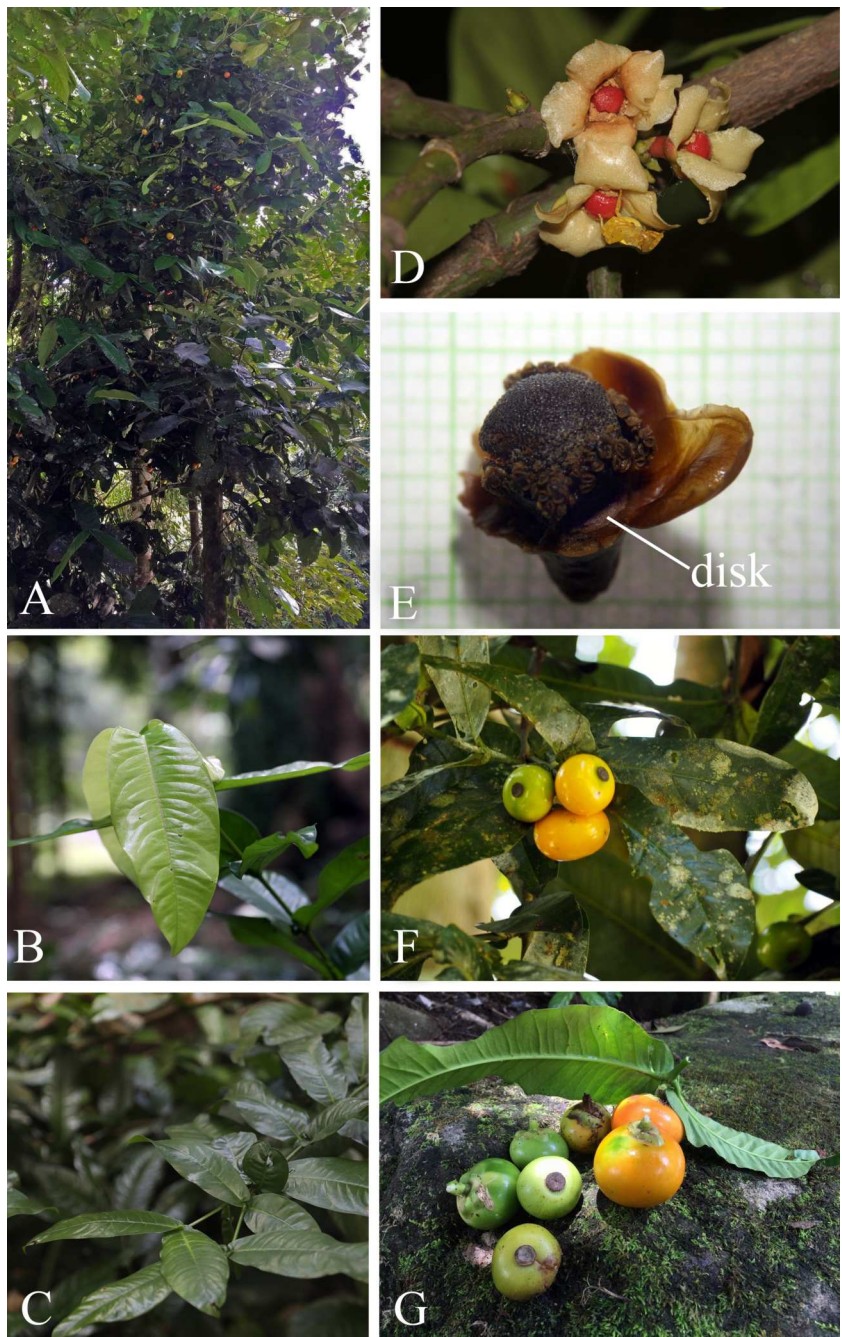

**Figure 15** *Garcinia prainiana.* (A) Habit. (B) Branchlets and young leaves. (C) Branchlets and mature leaves. (D) Inflorescences on short, leafless lateral branchlets with fully opened male flowers. (E) Male flower showing disk, stamen bundles, and pistillode (sepals and petals removed). (F–G) Branchlets, leaves, mature and ripe fruits. Photos: G Rawit Sichaikhan (A, F–G); Chatchai Ngernsaengsaruay (B–D); Pichet Chanton (E).

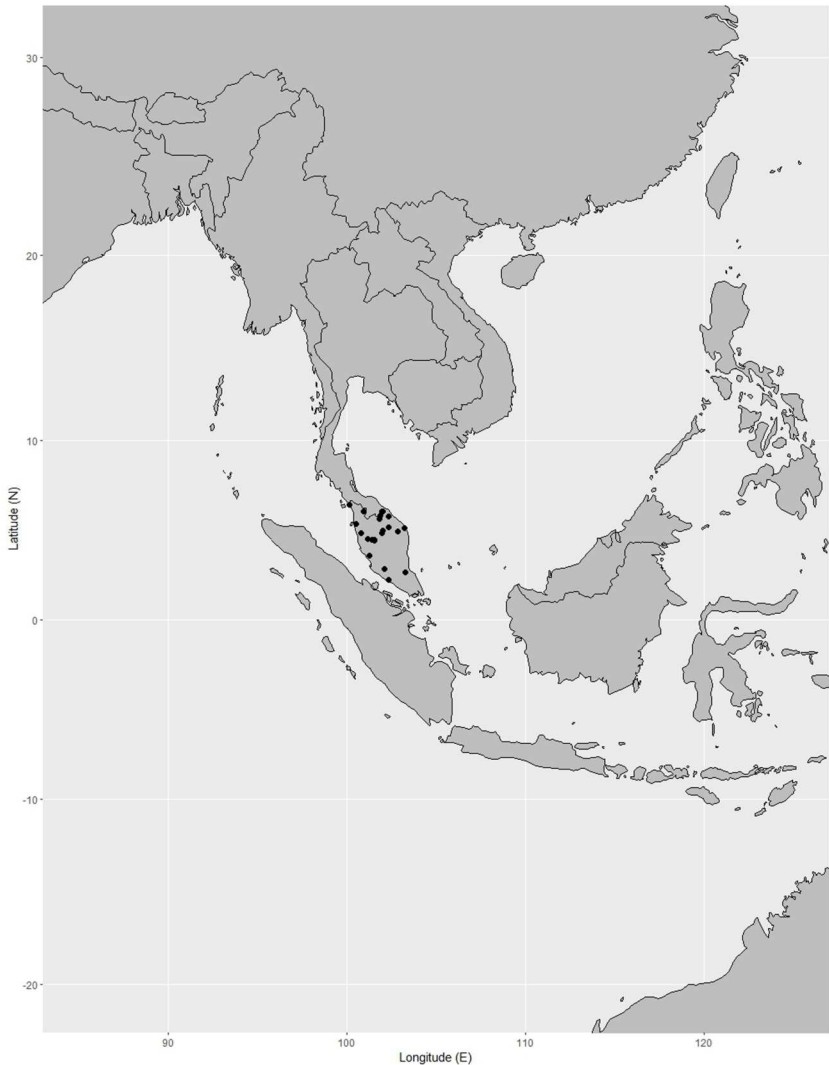

**Figure 16** **Distribution of *Garcinia prainiana*, known only from Peninsular Thailand and Peninsular Malaysia.** Map: Pichet Chanton & Chatchai Ngernsaengsaruay.

ornamental plant for use in landscape gardens in parks (*National Parks, Flora and Fauna Web, 2023*).

**Lectotypification.** *Garcinia prainiana* was named by *King (1890*: 171–172), who cited the specimen *Scortechini 1796* collected from Perak, "Kwala Dynong" but he did not mention the herbaria in which it was present. We located three sheets of the specimen *Scortechini 1796*: one sheet at CAL [CAL0000005844], one sheet at K [K000677678] and one sheet at P [P04701324], which were collected from the same locality, and following Art. 9.6 of the ICN (*Turland et al., 2018*), they constitute syntypes. Therefore, the CAL [CAL0000005844] specimen is selected here as the lectotype, following Art. 9.3 and 9.12 of the ICN (*Turland et al., 2018*).

**Additional specimens examined. Thailand. Central**: Nakhon Nayok [Phrueksaphan Thepparat Botanicical Garden, Chulachomklao Royal Military Academy, cultivated, 31 May 2019, *Ngernsaengsaruay & Boonthasak G30-31052019* (BK, BKF); **Peninsular**: Trang [Khao Chong Botanical Garden, Chong Subdistrict, Na Yong District, cultivated, 16 February 2022, *Ngernsaengsaruay, Meeprom & Boonthasak G32-16022022* (BK, BKF)], Yala [Chulabhorn Phatthana 7 Project, Than To District, near waterfall, 27 November 2019, *Ngernsaengsaruay & Sichaikhan G31-27112019* (BK, BKF)], Narathiwat [Hala-Bala Wildlife Sanctuary, Ban Bala, Lo Chut Subdistrict, Waeng District, 13 May 2005 [as *Garcinia* sp.], *Poopath 274* (BKF); Hala-Bala Wildlife Sanctuary, Waeng District, 22 September 2005, *Niyomdham & Puudjaa 7593* (BKF); Su-ngai Kolok District, 20 April 2002, *Upho 556* (QBG); Su-ngai Kolok District, cultivated, 20 May 2003, *Upho 550* (BKF)].

**Peninsular Malaysia**. Perak [Kwala Dipang, December 1896, *Curtis 3273* (K [K000677679]); Kg Kepayang near Ipoh, 30 October 1971, *Syed Abu Bakar Kep. FRI20440* (L [L2417220])], Pahang [Su-ngai Bertam at Kuala Mensum, 2 June 1971, *Whitmore Kep. FRI20091* (L [L2417222]); Path leading to Kuala Mensum from Boh Tea, Cameron Highlands, 24 September 1971, *Loh Kep. FRI19187* (L [L2417221]); Cameron Highlands Road, 18 January 1982, *Kochummen Kep. FRI29377* (L [L2417225]), Kelantan [0.5 mile east of Gua Musang, 14 July 1967, *Whitmore Kep. FRI4018* (L [L2417226]); Su-ngai Lebir, below Kuala Relai at Jentah, 24 April 1976, *Stone & Sidek 12426* (L [L2417224], BKF); Su-ngai Long off Su-ngai Pergau, Jeli, 26 September 1986, *Latiff et al. ALM1856* (L [L3806490], PSU); Ketam, Cicar Tinggi, Kampung Bata, Pasir Mas, 1 August 1992, *Noorsiha et al. Kep. FRI39214* (L [L3878683]); Pasir Putih, 23 October 1992, *Husmady et al. Kep. FRI39551* (L [L3806959]); near Brooke Camp, Gua Musang, 2 June 1994, *Husmady et al. Kep. FRI41841* (L [L2417223])].

*4. Garcinia xanthochymus* Hook. f. ex T. Anderson in Hook. f., Fl. Brit. India 1: 269. 1874; Kurz, J. Asiat. Soc. Bengal, Pt. 2, Nat. Hist. 43(2): 88. 1874 et Forest Fl. Burma 1: 93. 1877; Pierre, Fl. Forest. Cochinch. 1(5): 3. t. 71A. 1883; King, J. Asiat. Soc. Bengal, Pt. 2, Nat. Hist. 59(2): 168. 1890; Vesque, Epharmosis 2: 14. t. 82–84. 1889 et in A. DC. & C. DC., Monogr. Phan. 8: 315. 1893; Brandis, Indian Trees: 49. 1906; C. E. Parkinson, Forest Fl. Andaman Isl.: 89. 1923; Craib, Fl. Siam. 1(1): 118. 1925; Kanjilal, P. C. Kanjilal & A. Das, Fl. Assam 1(1): 104. 1934; Gagnep. in Gagnep., Fl. Indo-Chine Suppl.: 257. 1943; Maheshw., Bull. Bot. Surv. India 6: 114. t. 1. fig. 3. 1964; Whitmore in Whitmore, Tree Fl. Malaya 2: 222. 1973; C. J. Saldanha & Nicolson, Fl. Hassan Dist.: 127. 1976; Kosterm. in Dassan. & F. R. Forsberg, Revis. Handb. Fl. Ceylon 1: 87. 1980; D. G. Long in Grierson & D. G. Long, Fl. Bhutan 1(2): 370. fig. 30k–o. 1984; C. J. Saldanha & E. Rao, Fl. Karnataka 1: 207. 1984; P. H. Hô, Câyco Vietnam 1: 568. fig. 1573. 1991; E. W. M. Verheij & R. E. Coronel (eds.), PROSEA 2: 175, 176. 1992; N. P. Singh in B. D. Sharma & Sanjappa, Fl. Ind. 3: 129. 1993; S. Gardner, P. Sidisunthorn & V. Anusarnsunthorn, Field Guide Forest Trees of N. Thailand: 50. fig. 54. 2000; X. W. Li, J. Li, N. Robson & P. F. Stevens in C. Y. Wu, P. H. Raven & D. Y. Hong, Fl. China 13: 42. 2007; W. E. Cooper, Austrobaileya 9(1): 8. 2013; R. Tabassum, Angiospermic Fl. Gazipur Distr. Bangladesh (Dissertation): 110. 2015.—*Xanthochymus pictorius* Roxb., Pl. Coromandel 2(4): 51, t. 196. 1805, Hort. Bengal.: 42. 1814 et in Carey, Fl. Ind. 2: 633. 1832 [non *Garcinia pictoria* Roxb. (Roxburgh, 1832: 627–629)]; Dalzell &

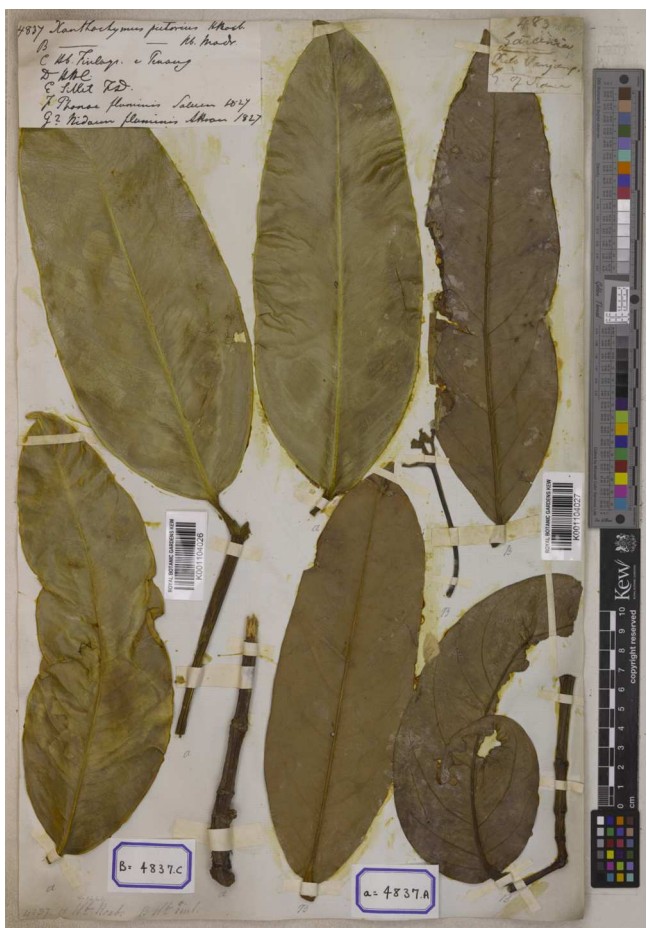

**Figure 17** **Lectotype of *Garcinia xanthochymus*, *Wallich Cat. 4837A* (K-W [K001104026]), Roxburgh's Herbarium.** Image: (c) RBG Kew, CC-BY 4.0.

A. Gibson, Bombay Fl.: 31. 1861.—*Stalagmitis pictoria* (Roxb.) G. Don, Gen. Hist. 1: 620. 1831.—*Garcinia pictoria* (Roxb.) Engl. in Engl. & Prantl, Die Naturlichen Pflanzenfamilien 3(6): 234. fig. 114 C–F. 1895, nom. illeg.—*Garcinia pictoria* (Roxb.) D'Arcy, Ann. Missouri Bot. Gard. 67: 998. fig. 4A. 1980, nom. illeg.—*Xanthochymus tinctorius* DC., Prodr. 1: 562. 1824, orth. var.—*Garcinia tinctoria* (DC.) W. Wight, Bull. Bur. Pl. Industr. U.S.D.A. 137: 50. 1909, orth. var.—*Garcinia tinctoria* Dunn in Gamble, Fl. Madras 1: 74. 1915, orth. var.—*Garcinia roxburghii* Kurz, Prelim. Rep. Forest Pegu App. A: xiii. 1875, nom. illeg. Type: Roxburgh's Herbarium, *Wallich Cat. 4837A* (lectotype designated here (K-W [K001104026, photo seen]) (Fig. 17).

    *Tree* evergreen, 10–30 m tall, 40–160 cm girth; latex white, turning creamish white, sticky; branches decussate, horizontal; branchlets green, 4-ridged, glabrous. *Bark* smooth or slightly rough, brown, dark brown or greyish brown; inner bark pale yellow. *Terminal bud* concealed between the bases of the uppermost pair of petioles. *Leaves* decussate; lamina variable in shape, oblong, elliptic, oblong-elliptic, narrowly oblong, narrowly elliptic or narrowly oblong-elliptic, sometimes ovate, 19–32.5 × 6–18 cm, apex acute or acuminate,

sometimes obtuse, base cuneate, margin repand and slightly revolute, thickly coriaceous, smooth (not bullate), shiny dark green above, paler below, glabrous on both surfaces, midrib and secondary veins flattened above, raised as a prominent ridge below, secondary veins 12–16 pairs, curving towards the margin connected in distinct loops and united into an intramarginal vein, conspicuous on both surfaces, with intersecondary veins, veinlets reticulate, visible on both surfaces, interrupted long wavy lines of differing lengths absent, sometimes indistinct below; petiole green, the uppermost pair of petioles reddish purple, turning green with age, 1.2–3.2 cm long, 2.5–7 mm in diam., not grooved, distinctly transversely rugose, glabrous, with a conspicuous basal appendage clasping the branchlet; young leaves shiny pale green; mature leaves turning greenish yellow to yellow before falling off. *Inflorescences* on short, leafless lateral branchlets, cymose, in lax fascicles of 3–8 flowers. *Flowers* unisexual, plants dioecious, 5-merous, petals erect with overlapping edges and corolla forming a bowl shape; bracteolate; sepals and petals quincuncial, coriaceous, concave. *Male flowers* the same size as in female flowers; bracteoles and pedicel same as in female flowers; sepals and petals same as or slightly smaller than in female flowers; disk in the center of the flower, intrastaminal, yellow, 5-lobed, fleshy, pitted, lobes positioned between the stamen bundles, antesepalous; stamens numerous, united in 5 bundles, 9–13 in each bundle, antepetalous, 0.8–1 cm long each bundle; filaments 0.7–2 mm long; anthers 2 thecae, 0.3–0.7 mm long; pistillode absent. *Female flowers* 1–1.3 cm in diam.; bracteoles triangular, 1.5–3 × 1.8–2.7 mm, apex acute, glabrous; pedicel green or reddish green, 1.5–5 cm long, widened at the apical part, middle part 1–2.5 mm in diam., apical part 2.5–4.5 mm in diam., terete, glabrous; sepals 5, pale green or greenish pale yellow, semi-orbicular or broadly obovate, 2.7–7 × 2.8–6.5 mm, unequal, apex rounded, margin ciliolate, glabrous outside; petals 5, whitish pale green or creamish white, suborbicular or broadly obovate, 0.7–1 × 0.7–1.2 cm, subequal, apex rounded, margin ciliolate; appendages 5, antesepalous, alternating with staminode bundles, fleshy, pitted; staminodes united in 5 bundles, 2–5 in each bundle, antepetalous, 2.2–4.2 mm long each bundle, whitish pale green or creamish white; pistil 0.5–1 cm long; ovary pale green, subglobose, globose or broadly ellipsoid, 3.2–7.5 × 3–6 mm, beaked, 1–3.5 × 1.3–2 mm (ovary including beaked looks like obpyriform in outline), unlobed, glabrous, 5-locular; stigma pale green, sessile, radiate, deeply 5-lobed, 4–7 mm in diam., papillate. *Fruits* berries, green, turning yellow when ripe, smooth, glabrous, glossy, with sticky yellow latex, then exocarp becoming dark brownish black and sinuously wrinkled when dry, subglobose, globose or broadly ovoid, 4.7–6 × 4–7.5 cm, sometimes oblique, asymmetrical, unlobed, with a short, thick beak, pericarp 1–1.5 cm thick, exocarp thin; persistent stigma dark brown or blackish brown, radiate, deeply 5-lobed; persistent sepals green, slightly larger than in flowering materials; fruiting stalk green or reddish green, 2.5–5 cm long, widened at the apical part, middle part 2.5–3.7 mm in diam., apical part 4–6 mm in diam., glabrous. *Seeds* 1–5, sometimes aborted, brown mottled with irregular lines, semi-ellipsoid, 1.8–3 × 1.2–2.7 cm, rounded at both ends, with yellow fleshy pulp. (Fig. 18).

**Distribution.** India, Andaman Islands, Nepal, Bhutan, Bangladesh, Myanmar, China, Vietnam, Thailand. This species has been introduced and is cultivated in many countries (*e.g.*, Panama, Africa, Sri Lanka, Peninsular Malaysia, Singapore, Queensland and French

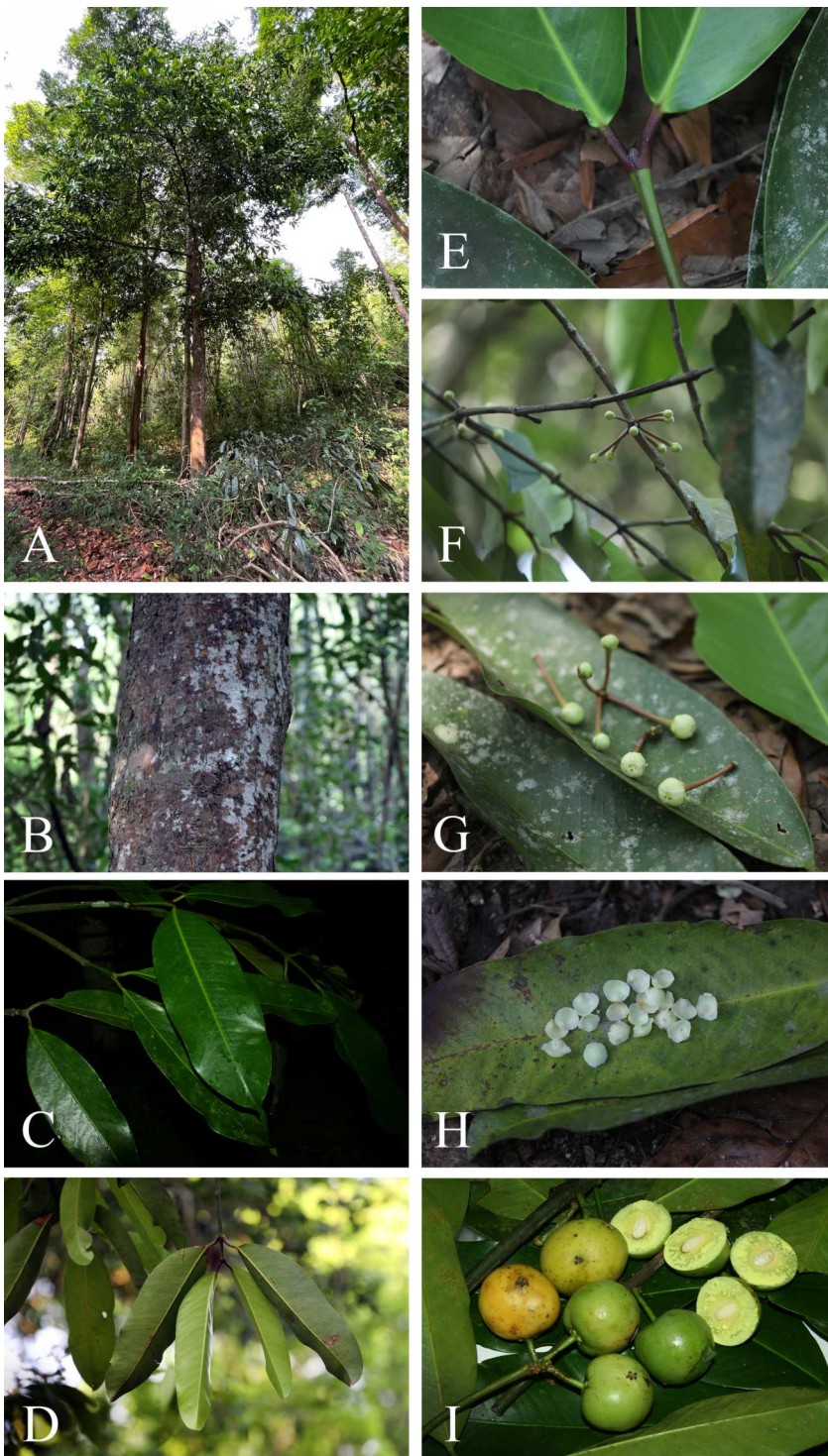

**Figure 18** *Garcinia xanthochymus.* (A) Habitat and habit. (B) Stem and outer bark. (C–D) Branchlets and leaves. (E) Terminal bud concealed between the bases of the uppermost pair of petioles. (F) Inflorescences on short, leafless lateral branchlets with young flower buds. (G) Inflorescence, fully opened female flowers and flower buds. (H) Fallen petals. (I) Fruiting branchlets with mature and ripe fruits, longitudinal sections of fruits with yellow latex and seeds. Photos: Chatchai Ngernsaengsaruay.

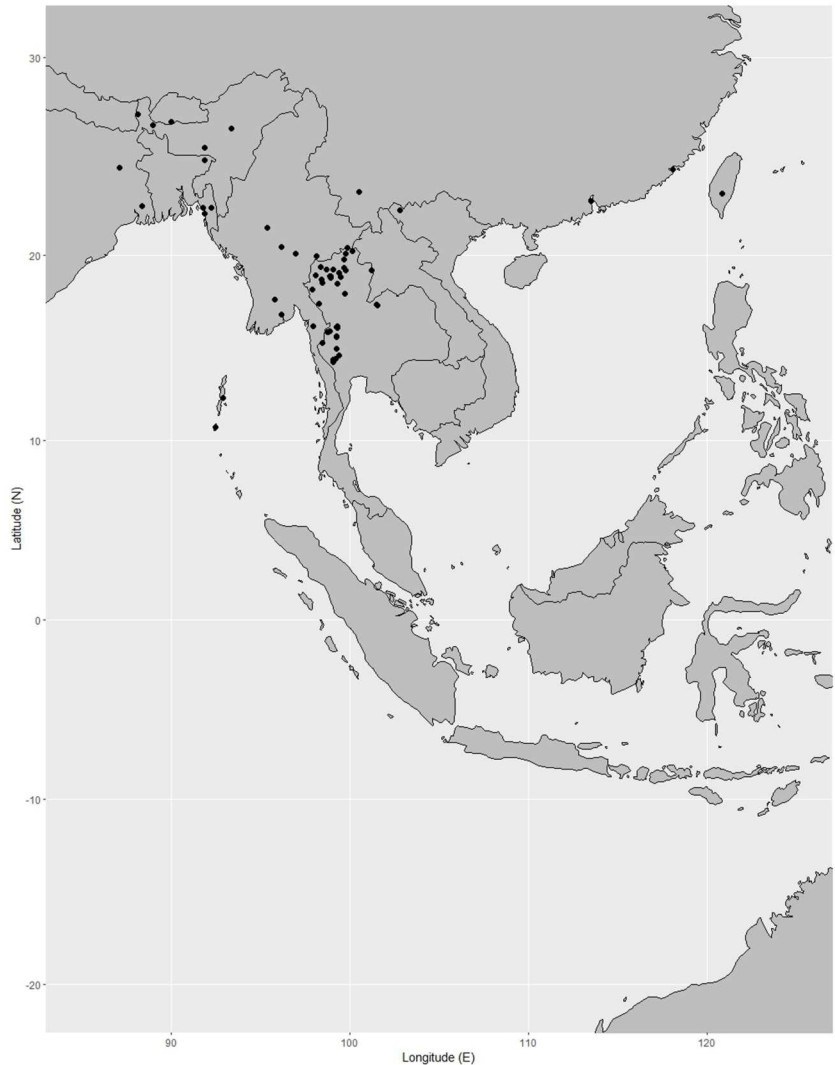

**Figure 19** **Distribution *Garcinia xanthochymus*, known from Indian subcontinent to Myanmar, China, Thailand and Vietnam.** Map: Pichet Chanton & Chatchai Ngernsaengsaruay.

Polynesia) (Fig. 19). It has been naturalized in Sri Lanka (*Kostermans, 1980*); from the specimens *Kostermans 24019*; *Jayasuriya 1626*; *Jayasuriya 1628*) and a few locations in Queensland (*Cooper, 2013*).

**Distribution in Thailand.** Northern: Mae Hong Son, Chiang Mai, Chiang Rai, Payao, Nan, Lampang, Phrae, Tak, Kamphaeng Phet; North-Eastern: Loei; South-Western: Uthai Thani, Kanchanaburi (Fig. 19).

**Habitat and Ecology.** It is found in dry evergreen, mixed deciduous and lower montane forests, sometimes in limestone areas, often along or near streams, 100–1300 m alt.

**Phenology.** Flowering and fruiting more than once, nearly throughout the year.

**Conservation Status.** *Garcinia xanthochymus* is widely distributed from Indian subcontinent to China and Indo-China. It is known from many localities and has a

large EOO of 3,579,093.97 km$^2$ and a relatively large AOO of 272 km$^2$. In Thailand, this species is known to be naturally distributed in the northern, the north-eastern and the south-western regions, and has an EOO of 165,710.42 km$^2$ and an AOO of 112 km$^2$. Because of this wide distribution and the number of localities, we consider the conservation assessment here as LC.

**Etymology.** The specific epithet of *Garcinia xanthochymus* is derived from the Greek compound words, *xantho-* meaning yellow, and *chymo*, meaning sap, referring to the plant with yellow latex (*Stearn, 1992*; *Radcliffe-Smith, 1998*; *Gledhill, 2002*).

**Vernacular Name.** Cha-kha-sa (จะคาสา) (Karen-Mae Hong Son); Mada (มะดะ) (Northern); Mada luang (มะดะหลวง) (Chiang Mai); Mapong (มะป้อง) (Phrae); Asam kandis (Malay); Dampel, Tamal (Hindi); Madaw (Myanmar); Rata goraka (Ceylon); Assam mangosteen, Egg tree, False mangosteen, Himalayan garcinia, Mysore gamboge, Sour mangosteen (English).

**Uses.** The fruits (pericarp and fleshy pulp) are edible and have a sour or sweet-sour taste and are used in the same way as *Garcinia dulcis*. *G. xanthochymus* is cultivated in Southeast Asia where the sour fruits can be used for making preserves, jams, sherbets, curries and vinegar. The latex from the bark and fruits provides a dye that is used in watercolor paints (*Maheshwari, 1964*; *Burkill et al., 1966*; *D'Arcy, 1980*; *Kostermans, 1980*; *Verheij & Coronel, 1992*; *Singh, 1993*). It is cultivated as an ornamental tree (*Maheshwari, 1964*; *D'Arcy, 1980*; *Singh, 1993*) and is suitable for streetscapes, parks and gardens (*National Parks, Flora and Fauna Web, 2023*). In India, the fruits are used for making medicaments (*Maheshwari, 1964*; *Burkill et al., 1966*) and the rootstocks are sometimes used for grafting on mangosteen (*G. mangostana* L.) (*Maheshwari, 1964*; *D'Arcy, 1980*; *Kostermans, 1980*; *Singh, 1993*). It contains many phytochemicals that can be extracted from its constituent parts: the bark, fruits, leaves, roots, twigs and seeds. The predominant extracted phytochemicals are xanthones, benzophenones, flavonoids, depsidones and isocoumarins. These phytochemicals contribute to the pharmacological activities of this plant as an antioxidant, antidiabetic and as having Nerve Growth Factor-potentiating, antimicrobial and cytotoxic activities. This species contains a broad range of phytochemicals with curative properties that can be greatly beneficial to man (*Che Hassan, Taher & Susanti, 2018*).

**Additional specimens examined. Thailand**. **Northern**: Mae Hong Son [Muang Soi Waterfall, Pai District, 17 January 1983, *Koyama et al. T-32653* (BKF); Kuet Luang Waterfall, Namtok Mae Surin National Park, Khun Yuam District, 23 May 2003 [as *Garcinia* sp.], *Wongprasert 035-50* (BKF, K, P [P06899613]); Huai Mae Sae, Salawin Wildlife Sanctuary, Mae Sariang District, 22 February 2007 [as *Garcinia* sp.], *Watthana 2302* (QBG)], Chiang Mai [Doi Suthep-Pui National Park, 23 May 1910, *Kerr 1201* (BM, K); ibid., 26 April 1958 [as *Garcinia* sp.], *Sørensen et al. 3059* (C); ibid., 29 April 1958 [as *Garcinia* sp.], *Sørensen et al. 3128* (C); ibid., 10 October 1958 [as *Garcinia* sp.], *Sørensen et al. 5568* (C); ibid., 28 November 1994, *Kopachon s003b1* (CMUB, L [L3813373]); ibid., 17 December 1995 [as *Garcinia* sp.], *BGO Staff 5455* (QBG); Ru Si Cave, Doi Suthep-Pui National Park, 5 January 1988, *Maxwell 88-1* (AAU, L [L2409261]); ibid., 13 April 2003, *Maxwell & Canines 2* (CMUB); Doi Suthep-Pui National Park, 2 December 2002, *Ngernsaengsaruay*

*G38-02122002* (BKF, dry and spirit materials); Mae Sa, Mae Rim District, 11 February 1923, *Kerr s.n.* (BM); Queen Sirikit Botanic Garden (originally called "Mae Sa Botanic Garden"), Mae Rim District, 9 December 1987, *Santisuk s.n.* (BKF); ibid., 3 July 1992, *Maxwell 92-361* (AAU, CMUB, E [E00160934], L [L3806732, L3806733], P [P04701167]); ibid., 15 October 1993, *Pooma 720* (BKF); ibid., 19 October 1993, *BGO Staff 033* (QBG); ibid., 5 March 1994, *BGO Staff 488* (QBG); ibid., 4 December 1998, *Prachit et al. 13* (QBG); ibid., 16 March 1999, *Panich 18* (PSU); ibid., 10 April 2007, *Glamwaewwong 1291* (QBG); Queen Sirikit Botanic Garden, Mae Rim District, 9 October 2003, *Ngernsaengsaruay G39-09102003* (BK, BKF, dry and spirit materials); Doi Chiang Dao, 10 January 1975 [as *Garcinia* sp.], *Geesink et al. 8213* (AAU, C, K, L [L2409576, L2409577], P [P05061711]); Chiang Dao District, 25 June 1989, *Maxwell 89-802* (L [L2409264]); Mae Pan, Doi Inthanon, 4 December 1969, *van Beusekom & Phengklai 2283* (AAU, BKF, E [E00839802], P [P05062058]); Ban Mai Phatthana, Nong Pa Ko Subdistrict, Mae Chaem District, 30 July 1988, *Maxwell 88-945* (BKF, L [L2409262, L2409263]); Ban Mae Mu, Mae Chaem District, 9 February 2017, *Pongamornkul 5951* (QBG); Pong Duet Hot Spring, Pa Pae Subdistrict, Mae Taeng District, 19 November 1992, *Maxwell 92-745* (CMUB, L (L3806731), P [P04701166])], Chiang Rai [Mae Chan District, 27 January 1970 [as *G. cambodgiensis*], *Sutheesorn 1550* (BK); Ban Dong Mada, Mae Lao District, 11 August 2007 [as *G. mckeaniana*], *Klomsakul 5* (BKF); Khun Chae National Park, 19 January 1998 [as *Garcinia* sp.], *Wongchai WU067* (BKF); Tham Luang-Khun Nam Nang Non Forest Park, Mae Sai District, 14 January 2011, *Norsaengsri & Tathana 7559* (QBG); ibid., 25 May 2011, *Norsaengsri & Tathana 7864* (BKF, QBG); Ban San Pa Sak, Tham Luang-Khun Nam Nang Non Forest Park, Mae Sai District, 29 March 2012, *Norsaengsri & Tathana 9297* (QBG)], Payao [Champathong Waterfall, Doi Luang National Park, 7 April 1999 [as *Garcinia* sp.], *Srisanga & Watthana 611* (QBG)], Nan [Sapan Waterfall, Khun Nan National Park, Bo Kluea District, 17 November 1993 [as *Garcinia* sp.], *Larsen et al. 44509* (AAU, BKF)], Lampang [Doi Khun Tan National Park, Hang Chat District, 2 October 1994, *Maxwell 94-1079* (BKF, CMUB, L [L3811034, L3811035]); Chae Son National Park, Mueang Pan District, 22 April 1996, *Maxwell 96-548* (BKF, CMUB, L [L3813120, L3813121]); Chae Son National Park, Mueang Pan District, 28 June 2002 [as *Garcinia* sp.], *Koonkhunthod et al. 204* (BKF); ibid., 28 June 2002, *Ngernsaengsaruay G36-28062002* (BK, BKF, dry and spirit materials); Wang Kaeo Waterfall (originally "Wahng Gayo Falls" on the label), Doi Luang National Park, Wang Nuea District, 27 March 1997, *Maxwell 97-274* (BKF, CMUB, L [L3883203])], Phrae [Huai Pu, 30 August 1982 [as *Garcinia* sp.], *Suvarnasuddhi 376* (BKF)], Tak [Doi Umphang, Umphang District, 9 May 1976 [as *Garcinia* sp.], *Sutheesorn 3756* (BK); Ti Lo Su Waterfall, Umphang Wildlife Sanctuary, Umphang District, 9 September 2002, *Ngernsaengsaruay 37-09092002* (BK, BKF, dry and spirit materials); Umphang Wildlife Sanctuary, Umphang District, 25 July 2008 [as *Garcinia* sp.], *Promhitathorn s.n.* (BKF); Tha Song Yang District, 29 May 2008, *Pooma et al. 7069* (BKF, L [L3811067])], Kamphaeng Phet [Huai Nam Lai, 8 November 1968, *Smitinand 10484* (BKF); ibid., 8 November 1968, *Smitinand 10486* (BKF); Khlong Lan Waterfall, Khlong Land District, 25 November 1997 [as *Garcinia* sp.], *Phengklai et al. 3918* (BKF, PSU)]; **North-Eastern**: Loei [Phu Luang Wildlife Sanctuary, 17 May 1998 [as *Garcinia* sp.], *Chayamarit et al. 1481* (BKF); Enroute from Lon Tae to Wang

Saphung, Phu Luang Wildlife Sanctuary, 17 May 1998 [as *Garcinia* sp.], *Wongprasert et al. s.n.* (BKF SN121739, the leaves belong to *G. xanthochymus* but the fruits belong to *Syzygium*; SN121740); Suan Hin Pha Ngam Park, Nong Hin District, 8 December 2014 [as *Garcinia* sp.], *Tagane et al. T3686* (BKF)]; **South-Western**: Uthai Thani [Huai Kha Khaeng Wildlife Sanctuary, Ban Rai District, 21 February 1970 [as *Garcinia* sp.], *van Beusekom & Santisuk 2895* (AAU, BKF, C, E [E00772042], L [L2409527, L2409528, L2409529], P [P05062005]); Huai Kha Khaeng Wildlife Sanctuary, 3 May 2023, *Ngernsaengsaruay G40-03052023* (BK, BKF); ibid., 3 May 2023, *Ngernsaengsaruay G40-03052023* (BK, BKF)], Kanchanaburi [Si Sawat District, 12 July 1962 [as *G. vilersiana*], *Suvarnakoses 2092* (BKF); Erawan National Park, between Khwae Noi and Mae Klong River, 17 April 1968, *van Beusekom & Phengklai 480* (AAU, C, E [E00839803], K, L [L2409581], P [P05062006]); Erawan Waterfall, Si Sawat District, 4 March 1975 [as *Garcinia* sp.], *Indrapong et al. 122* (BKF, C, K, L [L2409511, L2409512]); Khao Ko Kae, Erawan National Park, Si Sawat District, 3 November 1979 [as *G. nervosa*], *Shimizu et al. T-21656* (BKF); Erawan National Park, 7th waterfall, 2 September 1995 [as *Garcinia* sp.], *Parnell et al. 95-684* (K); Trail to Takhian Thong Waterfall, Sangkhla Buri District, 24 August 2010 [as *Garcinia* sp.], *Chamchumroon et al. 4786* (BKF)].

**India**. Sikkim, s.d. [as *X. pictorius*], *Hooker s.n.* (E [E00839669], P [P04701144]); Assam, s.d. [as *X. pictorius*], *Masters s.n.* (L [L2409240], P [P04701146, P04701149]); Khasia, s.d. [as *X. pictorius* ], *Hooker & Thomson s.n.* (P [P04701148, P04701153], G [G00458423]); Shillong, Meghalaya, 20 August 1886, *Clarke 44626* (G [G00458424]); Neilgherries, 1857 [as *X. pictorius*], *Cleghorn s.n.* (E [E00839668]); Malabar Concan, s.d. [as *X. pictorius*], *Stocks & Law s.n.* (P [P04701151], G [G00458422]); East Bengal, s.d. [as *X. pictorius*], *Herb. Griffith 872* (P [P04701147]); Peninsula Indiae Orientalis, distributed at the Royal Gardens, Kew, 1866–1867 [as *X. pictorius*], *Herb. Wight 139* (L [L2409239]); Flora of the Andaman Islands, 1884, *King's Collector 242* (L [L2409235], P [P04701159]); Guitar Island, Andaman Islands, February–March 1934, *Ram 3689* (E [E00839676]); cultivated in Hort. Bot. Calcuttensis, 1864, *Pierre 3385* (P [P04701152, P04701160, P04701161]); ibid., s.d. [as *X. pictorius*], *Unknown s.n.* (L [L2409247], P [P04701154]); H.B.C. (Calcutta Botanical Garden), 1832 [as *X. pictorius*], *Unknown Cat. 4837D* (EICH 4837D) (K-W [K001104029], G [G00458427]).

**Sri Lanka**. Udawatte Kele above Kandy, naturalized, 7 May 1971, *Kostermans 24019* (C, L [L2409230, L2409231, L2409232], P [P04701162, P04701163]); Kandy town, June 1971, *Kostermans s.n.* (L [L2409241]); Udawatte Kele Sanctuary, Kandy District, naturalized, 30 April 1974, *Jayasuriya 1626* (L [L2409548]); ibid., naturalized, *Jayasuriya 1628* (L [L2409549]); locality not specified, s.d., *Walker 128* (E [E00839683]).

**Bhutan**. Longa Khola near Phipsoo, Sarbhang District, 17 March 1982, *Grierson & Long 3804* (E [E00170193]).

**Bangladesh**. Sylhet (originally ''Sillet'' on the label), April 1824 [as *X. pictorius*], *de Silva Cat. 4837E* [East India Company Herbarium (EICH) 4837E] (G [G00458431], K-W [K001104030], P [P04701157]); Flora of the Chittagong Hill Tracts, 1886, *King's Collector 421* (E [E00839677], L [L2409237], P [P04701150]); Chittagong, s.d. [as *X. pictorius*], *Hooker & Thomson s.n.* (P [P04701145]); E. Bengal, 10 May 1945, *Sinclair 4273*

(E [E00839671]); Kaptai, Sitapahar west, Jarultala (Silchari), Rangamati District, 24 April 1997, *Huq & A. I. 10466* (L [L3883042, L4449015]).

**Myanmar**. Phanoe hills, Salween River (originally "Phanoe fluminis Saluen" on the label), 13 March 1827 [as *X. pictorius*], *Wallich Cat. 4837F* (EICH 4837F) (G [G00458466], K-W [K001104031]); Nidaun, Ataran River (originally "Nidaun fluminis Attran" on the label), 28 January 1827 [as *X. pictorius*], *Wallich Cat. 4837G* (EICH 4837G) (G [G00458433], K-W [K001104032]); ibid. (originally "Nidong on the Attran" on the label), s.d., *Wallich 1903* (G [G00458428]); Shan, October 1891, *Huk 57* (P [P04701156]); Phanse on the Saluen, s.d. [as *X. tinctorius*], *Wallich 1902* (G [G00458455]); Gamon Reserve, Tharrawaddy District, Bago, 9 May 1912, *Smales 46* (E [E00839670]); Myingyan District, 10 August 1914, cultivated, *Rogers 410* (E [E00839684]); Pozaungdaung Reserve Compart. 8, Yamethin District, Mandalay, 9 January 1915, *Rogers 556* (E [E00839672]); Yangon (originally "Rangoon" on the label), March 1938, *Dickason 7142* (E [E00839682]; L [L2409234]).

**China**. Mienning, Mayetui, Yunnan, 24 September 1938, *Yu 17709* (E [E00839463]); Guangzhou, 1982 [as *G. tinctoria*], *Xing et al. 166* (E [E00839462]); Chia I Agriculture Research Institute, Taiwan, 18 May 1966, *Liao 10585* (L [L2409233, L2409243]); Fujian, Xiamen, 27 May 2020, *Wang Tao 175* (AU [AU080688]).

**Country not specified**. Locality not specified, s.d. [as *X. pictorius*], *Herb. Roxburgh, Unknown Cat. 4837A* (EICH 4837A) (K-W [K001104026]); locality not specified, 1832 [as *X. pictorius*], Herb. Madras, *Unknown Cat. 4837B* (EICH 4837B) (K-W [K001104028], G [G00458430]).

## DISCUSSION

According to *Anderson (1874)*, *Vesque (1893)*, and *Jones (1980)*, *Garcinia* sect. *Xanthochymus* is distinguished from other sections by its pentamerous flowers (rarely tetramerous); however, all species examined in this paper, have pentamerous flowers.

All Thai species of *Garcinia* sect. *Xanthochymus* have male flowers that have a fertile whorl of stamens that united into 5 bundles (also called phalanges), and these are antepetalous (opposite petals). The whorl of stamen bundles surrounds a lobed structure in the center of the flower, called "intrastaminal disk" that can be found in *G. dulcis*, *G. nervosa*, and *G. xanthochymus* except *G. prainiana* has a small ring-shaped disk surrounding the base of the pistillode (Fig. 14E). The female flowers have a whorl of staminodes that are united into 5 bundles, alternating with fleshy antepetalous structures, called "intrastaminal appendages" (also called "disk lobes") that can be found in *G. dulcis*, *G. nervosa*, and *G. xanthochymus* except *G. prainiana* (staminodes absent). The botanical terminologies "disk" and "appendages" follow those of *Sweeney (2010)*.

As stated previously, many morphological features and molecular data do not support the placement of *Garcinia prainiana* into *G.* sect. *Xanthochymus*, but rather support its placement into a clade that includes members of *Jones (1980)* sections *Mungotia*, *Macrostigma*, and *Tripetalum* (part of Lineage B in *Sweeney, 2008*). Species in these sections do not have "disks" like those in *G.* sections *Xanthochyms*, *Rheedia*, *Teracentrum*, or

*Rheediopsis* (Lineage A in (*Sweeney, 2008*), see also (*Sweeney, 2010*)). However, from our observations we found a small disk ring-shaped surrounding the base of the pistillode of *G. prainiana* (Fig. 14E). Therefore, it is placed into *G.* sect. *Xanthochymus* follow those of *Vesque (1893)* and *Jones (1980)*.

*Kochummen & Whitmore (1973)* mentioned *Garcinia* sect. *Xanthochymus* has a small central pistillode, but from our observations, the pistillode can be very small or absent: *G. dulcis* (present or absent), *G. nervosa* and *G. xanthochymus* (absent), and *G. prainiana* (present). According to *Whitmore (1973)*, the pistillode in male flowers of *G. nervosa* is tiny; however, from our examination of specimens, the pistillode is absent.

The terms aril, pulp, and pulpy aril are commonly used in *Garcinia* (*Anderson, 1874*; *Ridley, 1922*; *Whitmore, 1973*; *Jones, 1980*; *Singh, 1993*). The aril is an outgrowth of the funicle (funiculus), forming an appendage enveloping the seed (*Hickey & King, 2000*; *Beentje, 2010*), while the sarcotesta is a fleshy layer surrounding the seed and develops from the outer seed coat (*Beentje, 2010*). *Corner (1976)* studied the anatomy and morphology of *Garcinia* (and other Clusiaceae) fruits and seeds, publishing the results in ''*The Seeds of Dicotyledons*''. He interpreted the pulpy material that surrounds the seeds of species of *Garcinia* as endocarp. The testa was described as exarillate and unspecalized, except for secretory canals. For *G. xanthochymus* he described the testa as being 1–2 mm thick, many cells thick, undifferentiated, permeated by a close reticulum of vascular bundles and the endocarp as composed of tangentially elongate, collenchymatous cells, without secretory canals. From our examination of specimens, we suggest using fleshy pulp.

According to previous studies (*e.g.*, *Pierre, 1883*; *Vesque, 1889*; *King, 1890*; *Vesque, 1893*; *Pitard, 1910*; *Ridley, 1922*; *Maheshwari, 1964*; *Whitmore, 1973*; *Singh, 1993*) and based on the specimens that we examined *Garcinia dulcis*, *G. vilersiana*, *G. cambodgiensis*, *G. andamanica*, and *G. andamanica* var. *pubescens* are very closely related and several characters of the habit, leaves, flowers, and fruits overlap among them. The indumentum density of the branchlets, leaves, petioles, sepals, and pedicels vary from pubescent to glabrescent and sometimes glabrous. Therefore, four taxa, namely *G. andamanica*, *G. andamanica* var. *pubescens*, *G. cambodgiensis*, and *G. vilersiana* are not morphologically distinguishable from *G. dulcis* and are treated here as new synonyms.

*Vesque (1889)*, *Vesque (1893)*, and *Pitard (1910)* described characters of *Garcinia cambodgiensis* as: tree, 8–12 m tall; branchlets 4-angled; leaves variable in shape, oblong, elliptic-oblong, lanceolate, ovate or oblong-ovate, 25–37 × 7–18 cm, coriaceous; petiole 2–4 cm long, glabrous or hairy; flower 5-merous; pedicel 1.5–3 cm long; ovary 5-locular; stigma 5-lobed; fruits globose, subglobose or depressed globose, 3–4 × 3.5–4.5 cm; seeds 2–5. Based on these characters, *G. cambodgiensis* is not morphologically distinguishable from *G. dulcis*, and is synonymized under *G. dulcis*.

According to previous studies (*e.g.*, *Pierre, 1883*; *Vesque, 1889*; *King, 1890*; *Vesque, 1893*; *Ridley, 1922*; *Whitmore, 1973*; *Singh, 1993*) and based on the specimens that we examined, *Garcinia nervosa*, *G. nervosa* var. *pubescens*, and *G. spectabilis* are very closely related and several characters of the habit, leaves, flowers, and fruits are overlap between the three taxa. The indumentum density of the branchlets, leaves, and petioles vary from pubescent to glabrescent. Therefore, two taxa, *i.e.*, *G. nervosa* var. *pubescens* and *G. spectabilis* cannot

be distinguished from *G. nervosa* and are synonymized under *G. nervosa* in this taxonomic treatment.

*Garcinia dulcis* and *G. xanthochymus* are very morphologically similar and closely related and many previous authors have had difficulty distinguishing between these two species. Our view based on the specimens that we examined is that *G. dulcis* and *G. xanthochymus* are very closely related and several characters of the habit, leaves, flowers, and fruits are overlap between the two species, but *G. dulcis* differs in having its inflorescences in dense fascicles of flowers (*vs* in lax fascicles of flowers); leaves slightly bullate to bullate (*vs* smooth or plane, not bullate); branchlets, leaves, petioles, sepals, and pedicels mostly pubescent to glabrescent (*vs* glabrous); and plants polygamo-dioecious (*vs* dioecious). In addition, leaf shapes of *G. dulcis* and *G. xanthochymus* are variable and overlap between species. The shapes of laminas of *G. dulcis* are mostly lanceolate-ovate or lanceolate (usually broadest at the basal part and gradually narrowing towards the apex) and are sometimes narrowly oblong, oblong or elliptic (usually broadest at the middle part but not gradually narrowing towards the apex), while the shapes of laminas of *G. xanthochymus* can be oblong, elliptic, oblong-elliptic, narrowly oblong, narrowly elliptic or narrowly oblong-elliptic (usually widest at the middle part but not gradually narrowing towards the apex) and are sometimes ovate (usually widest at the basal part and gradually narrowing towards the apex).

*Garcinia dulcis* is similar to *G. xanthochymus* and is distinguished by its shorter pedicels (0.6–1 cm long) and flowers almost closed. In contrast, *G. xanthochymus* has longer pedicels (c. 2.5 cm long) and expanded flowers (*Kurz, 1874*; *Kurz, 1877*; *Maheshwari, 1964*). From our observations, we found the pedicel of *G. dulcis* is 0.5–2 cm long and the pedicel of *G. xanthochymus* is 1.5–5 cm long, while the corolla of both species forming a bowl-shaped and the petals are erect with overlapping edges. Therefore, we cannot be separated between species by overlapped characters.

*Garcinia dulcis* is related to *G. xanthochymus*, but differs in having its secondary veins 9–14 pairs (*vs* 20–25 pairs) (*Cooper, 2013*); however, from our study, we found the numbers of secondary veins of *G. dulcis* (10–17 pairs) and *G. xanthochymus* (12–16 pairs) are overlap, which cannot be distinguished between species.

*Kurz (1877)* reported all parts of *Garcinia dulcis* are glabrous, but from the author's observations, we found the indumentum density of several parts (branchlets, leaves, petioles, sepals, and pedicels) vary from pubescent to glabrescent and sometimes glabrous.

*Maheshwari (1964)* notes that *G arcinia xanthochyms* has ciliate sepal tips and that *G. dulcis* does not. However, from our examination of specimens, these two species have ciliolate sepal and petal margins.

According to previous studies of *Garcinia nervosa*, the shape and size of leaves are oblong, narrowly oblong, oblong-ovate or oblong-lanceolate and 22–60 × 9–25 cm (*Ridley, 1922*; *Corner, 1952*; *Whitmore, 1973*; *Keng, 1990*). Furthermore, from our examinations, we found the laminas of this species are more in shape, narrowly elliptic, narrowly oblong, oblong, lanceolate, lanceolate-ovate or elliptic-oblong, and larger 33.5–80 × 8–27 cm and has the largest leaves in *Garcinia*. According to previous studies, the numbers of secondary veins are 15–20 pairs (*Ridley, 1922*) and the length of petioles is 2.5–3.8 cm long (*Ridley, 1922*; *Corner, 1952*; *Whitmore, 1973*); however, from our investigations, we found the

secondary veins are sometimes more in numbers, 11–24 pairs and the petioles are often longer, 2.3–7.2 cm long.

*Garcinia nervosa* was recorded for the first time from Assam in mainland India (*Dutta et al., 2014*). We have seen photos of leaves, female flowers, fruits and seeds in this paper and all characters resemble *G. dulcis*. We have not seen the specimens from Assam before and with a disjunct distribution between Assam and Peninsular Thailand to the Malesian region. Therefore, we have doubts that the materials from Assam belong to the same species as *G. nervosa*.

*Garcinia prainiana* is similar to *G. phuongmaiensis* in having coriaceous, bullate, shiny dark green, subcordate, subamplexicaul leaves with a short petiole, but differs in relatively larger habit, tree, 3–12 m tall (*vs* shrub, 1–3 m tall); larger leaves, 12.5–27.5 × 5.5–11.5 cm (*vs* 4–11 × 2.5–5 cm); larger flowers, 2.5–4 cm in diam. (*vs* c. one cm in diam.); variable in petal color (see description) (*vs* white petals); staminodes absent (*vs* present); unlobed fruits, bright yellow, orangish yellow and bright orange when ripe (*vs* shallowly 3–4-lobed fruits, bright red when ripe); seeds with pale orange fleshy pulp (*vs* seeds with white fleshy pulp); and it is distributed in Peninsular Malaysia and Peninsular Thailand (*vs* central Vietnam). The characteristics and distribution of *G. phuongmaiensis* were taken from *Tuan et al. (2023)*.

*G. phuongmaiensis* is also similar to *G. nuntasaenii* in its habit (shrub), white latex, turning pale yellow, coriaceous, shiny dark green, subcordate leaves with a short petiole, flowers c. one cm in diam., pale yellow or creamy white petals, and fruits turning red when ripe, but differs in having 5-merous flowers (*vs* 4-merous flowers) and 3–4-lobed fruits (*vs* 4–6-lobed fruits). (*Ngernsaengsaruay & Suddee, 2016*; *Tuan et al., 2023*).

In Peninsular Malaysia, *Garcinia prainiana* grows up to c. 18 m tall (*Corner, 1952*; *Allen, 1965*) and can reach 20 m tall (from the specimen *Whitmore Kep. FRI4018*), but from our field observations and examination of specimens, it is a small to medium-sized tree, usually grows 3–12 m tall.

According to *Whitmore (1973)* and *Syazwani (2020)*, the shapes of leaves of *Garcinia prainiana* are ovate-oblong or ovate; however, in this study, we found the leaves can be elliptic, oblong or elliptic-oblong, sometimes narrowly oblong, which are in consistent with *King (1890)*, *Ridley (1922)*, *Corner (1952)*, and *Allen (1965)*.

According to *Zawiah & Othaman (2012)*, in Peninsular Malaysia, the young leaves of *Garcinia prainiana* are reddish; however, from our observations, we found the young leaves can be pale green in agreement with *Corner (1952)* and *Allen (1965)*.

*Xanthochymus pictorius* was named by *Roxburgh (1805*: 51–52. t. 196), who mentioned that the species is a native of moist valleys among the Circar mountains. *Anderson (1874*: 269) listed *X. pictorius* Roxb. and *X. tinctorius* DC. as synonyms of *Garcinia xanthochymus* Hook. f. ex T. Anderson, together with the specimen *Wallich Cat. 4837*. This specimen represents seven different materials collected from seven different localities, which are distinguished by A, B, C, D, E, F, and G, respectively. The specimen *Wallich Cat. 4837A* (K-W [K001104026]) is from Roxburgh's Herbarium, *Wallich Cat. 4837B* (K-W [K001104028]) is from Madras Herbarium, *Wallich Cat. 4837C* (K-W [K001104027]) is from Penang, Finlayson's Herbarium, *Wallich Cat. 4837D* (K-W [K001104029]) is

from Calcutta Botanical Garden (H.B.C.), *Wallich Cat. 4837E* (K-W [K001104030]) is from Sylhet collected by F. De Silva, *Wallich Cat. 4837F* (K-W [K001104031]) is from Phanoe Hills, on the Salween River collected by N. Wallich, and *Wallich Cat. 4837G* (K-W [K001104032]) is from Nidaun, Ataran River collected by N. Wallich. Following advice in *Forman (1997)*, the specimen *Wallich Cat. 4837A* (K-W [K001104026]) should be considered as a lectotype.

*Garcinia xanthochymus* was named by *Anderson (1874)* and is the correct name for the species under *Garcinia*. However, nomenclature for this species commenced by *Roxburgh (1805)*, who described *Xanthochymus pictorius* [non *G. pictoria* Roxb.]. The name *X. tinctorius* published in *de Candolle (1824)* was a misspelling for *X. pictorius*, hence the new combinations under *Garcinia* were superfluous including *G. pictoria* (Roxb.) Engl. (*Engler, 1895*) and *G. pictoria* (Roxb.) (*D'Arcy, 1980*) and were based on erroneous interpretation.

## CONCLUSIONS

Investigations of *Garcinia* sect. *Xanthochymus* in Thailand, especially those dealing with taxonomy and morphology, have for a long time been complicated by the scarcity of information about morphological characteristics of vegetative and reproductive parts of representative species. We have clarified with updated morphological descriptions and an identification key for four species of *G.* sect. *Xanthochymus* (*G. dulcis*, *G. nervosa*, *G. prainiana*, and *G. xanthochymus*) in Thailand.

Four taxa, *Garcinia andamanica*, *G. andamanica* var. *pubescens*, *G. cambodgiensis*, and *G. vilersiana* cannot be distinguished from each other and are indistinguishable from *G. dulcis*, and are synonymized under *G. dulcis*. Two taxa, *G. nervosa* var. *pubescens* and *G. spectabilis* are indistinguishable from *G. nervosa* and are treated here as new synonyms.

Nine names in *Garcinia* sect. *Xanthochymus* are lectotypified here: *G. dulcis* and its associated synonyms (*G. cambodgiensis* and *G. vilersiana*), *G. nervosa* and its associated synonyms (*G. andersonii*, *G. nervosa* var. *pubescens*, and *G. spectabilis*), *G. prainiana*, and *G. xanthochymus*.

*Garcinia* sect. *Xanthochymus* in Thailand is distinguished from other sections by its pentamerous flowers; sepals and petals with quincuncial aestivation; numerous stamens united into 5 bundles; and fleshy fruit with a thin exocarp, usually sinuously wrinkled when dry.

In Thailand, two species, *Garcinia prainiana* and *G. nervosa* are confined to the peninsular region and the other two species, *G. dulcis* and *G. xanthochymus*, have a wider distribution.

*G. prainiana* is uncommon in Thailand, only known from two provinces (Yala and Narathiwat), and is widely distributed in Peninsular Malaysia. It is found in tropical lowland evergreen rain forests, occasionally along streams, at elevations of 30–200 m amsl. *G. nervosa* is known from seven provinces (Nakhon Si Thammarat, Phatthalung, Trang, Satun, Pattani, Yala, and Narathiwat) and has a wide distribution in Andaman and Nicobar Islands to the Malesian region. It usually grows in tropical lowland evergreen rain forests,

sometimes in limestone areas, often near or along streams, at elevations of 50–250(–600) m amsl. *G. xanthochymus* is found in three Thailand floristic regions, the northern, the north-eastern, and the south-western and is widely distributed from Indian subcontinent to Myanmar, China, and Vietnam. The habitat preference of the species is dry evergreen, mixed deciduous, and lower montane forests, sometimes in limestone areas, often along or near streams, at elevations of 100–1300 m amsl. *G. dulcis* occurs in four Thailand floristic regions, the eastern, the south-western, the south-eastern, and the peninsular regions and is very widely distributed from India (including Andaman Islands), Indo-China, the Malesian region to Australia (Queensland) and French Polynesia. It is found in dry evergreen, tropical lowland evergreen rain forests, and lower montane forests, often in limestone areas, sometimes along streams, at elevations of 0–1100 m amsl. All species have a conservation assessment of Least Concern.

The fruits of all species are edible and have a sour or sweet-sour taste. *G. dulcis* is often cultivated as a fruit tree in all floristic regions of Thailand and *G. prainiana* is locally cultivated for its fruits in southern Thailand. Several previous studies reported all species contain many phytochemicals, which have potential in the development of pharmaceutical products.

## ACKNOWLEDGEMENTS

We would like to thank the curators and staff of the following herbaria AAU, BK, BKF, BM, C, CMUB, K, P, PSU, QBG, and SING for their assistance during visits and allowing access to the herbarium specimens, and those included in the digital herbarium databases of AAU, AU, BM, BR, CAL, E, G, K (including K-W), L (including U), P, and US. We are grateful to the plant collectors of *Garcinia* sect. *Xanthochymus*. We also would like to thank Wanwisa Bhuchaisri for the line drawings, G Rawit Sichaikhan for the specimens and photos of *G. prainiana* from Yala Province, Nattanon Meeprom, Weereesa Boonthasak, and Paweena Wessapak for their kind help with field work.

### Funding

This research was funded by the Basic Research Fund (BRF) and the International SciKU Branding (ISB), Faculty of Science, Kasetsart University. The funders had no role in study design, data collection and analysis, decision to publish, or preparation of the manuscript.

### Grant Disclosures

The following grant information was disclosed by the authors:
The Basic Research Fund (BRF) and the International SciKU Branding (ISB), Faculty of Science, Kasetsart University.

### Competing Interests

The authors declare there are no competing interests.

## Author Contributions

- Chatchai Ngernsaengsaruay conceived and designed the experiments, performed the experiments, analyzed the data, prepared figures and/or tables, authored or reviewed drafts of the article, and approved the final draft.
- Pichet Chanton performed the experiments, prepared figures and/or tables, authored or reviewed drafts of the article, and approved the final draft.
- Minta Chaiprasongsuk performed the experiments, authored or reviewed drafts of the article, and approved the final draft.
- Nisa Leksungnoen performed the experiments, authored or reviewed drafts of the article, and approved the final draft.

## Field Study Permissions

The following information was supplied relating to field study approvals (i.e., approving body and any reference numbers):

Department of National Parks, Wildlife and Plant Conservation

## Data Availability

The raw data is described in the Additional specimens examined.

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
