# Peer review of "A taxonomic revision of Garcinia section Xanthochymus (Clusiaceae) in Thailand"

_PeerJ, doi:10.7717/peerj.16572_

## Round 0.1 · original submission · Major Revisions

I completely agree with the two reviewers that raised concerns on nomenclatural and taxonomical issues as well as the lack of clarity of the attributes separating species in the Garcinia section Xanthochymus. In addition, check whether G. prainiana belongs to this section, as one of the reviewers mentioned or justify the incusion of this species into the section. In all cases review the attributes utilized to recognized the species in this section. Before returning the manuscript check carefully the English of the manuscript.

**Language Note:** The Academic Editor has identified that the English language must be improved. PeerJ can provide language editing services - please contact us at copyediting@peerj.com for pricing (be sure to provide your manuscript number and title). Alternatively, you should make your own arrangements to improve the language quality and provide details in your response letter. – PeerJ Staff

Reviewer 1 ·

Basic reporting

No comment

Experimental design

No comment

Validity of the findings

No comment

Additional comments

Dear Authors

The manuscript is well prepared by the authors, however it needs minor revisions and careful examination of it before it is published.

Annotated reviews are not available for download in order to protect the identity of reviewers who chose to remain anonymous.

Reviewer 2 ·

Basic reporting

This paper aims to re-evaluate the taxonomy of Garcinia section Xanthochymus in Thailand, ultimately determining the recognition of four distinct species through the processes of synonymizations and lectotypifications. The manuscript is well written, and I generally agree with their taxonomic treatments. However, I do have reservations regarding the decision to synonymize G. cambodgiensis with G. dulcis. Based on my limited understanding, potential distinguishing characteristics include petiole length, the presence of petiole hair, and the number of secondary veins. Additionally, G. cambodgiensis reportedly exhibits depressed fruits at the apex as illustrated in Flore Forestière de la Cochinchine (pl. 72), differing from a short beak at the apex as described by the authors. While these variations could potentially be attributed to polymorphism within the species, it remains crucial for these distinctions to be clarified within the manuscript. I am of the opinion that this manuscript could significantly enhance its quality by addressing the aforementioned points. Therefore, I strongly recommend that the authors consider making revisions to the manuscript.

Issues that need to be addressed:
1. Elaborate on the rationale behind the decision to synonymize G. cambodgiensis with G. dulcis.

Other comments
L35, L68. Some species shows trioecious. See “Joseph, K. S. and H. N. Murthy (2015) Sexual system of Garcinia indica Choisy: geographic variation in trioecy and sexual dimorphism in floral traits. Plant Systematics and Evolution 301(3): 1065–1071.”
L64. Change italic to regular “Ngernsaengsaruay et al. (2022)”
L131. Broken link to GeoCAT (http://www.geocat.kew.org)
L132. Is “ที่ ทส“ needed?
L.281. In Flore Générale de l'Indo-Chine, fruits of G. cambodgiensis have “deprimee legerement entre les graines, a 2-5 loges”.
L308. ducis -> dulcis
L318. Change “Maphut” from bold to regular font
L632. km2 -> km2
L641. Change “Maphut pa” from bold to regular font
L824. Change “Chupu” from bold to regular font
L961. Change “Mada luang” from bold to regular font
L1125, L1128. Please provide clarification regarding the fundamental characteristics.
L1250-. It is advisable to standardize the usage of “above mean sea level” and “amsl.”

Experimental design

no comment

Validity of the findings

no comment

Reviewer 3 ·

Basic reporting

Generally, the paper was well written with adequate background/context provided and relevant literature cited . There were a few grammatical issue scattered throughout, and I have suggested corrections/edits in a marked-up version of the manuscript that was uploaded as part of the review.

Experimental design

The material presented in the manuscript is within scope for the journal and is on par with other similar taxonomic papers submitted to PeerJ.

Validity of the findings

Generally the validity of the findings are robust. However, before the manuscript is ready for publication there are a few notable instances where the authors need to provide clarification or additional evidence to support their case.

1) Molecular data (Sweeney 2008) and morphology do not support the placement of G. prainiana into Sect. Xanthochymus. This species belongs to a clade of species that Jones (1980) placed into sects. Mungotia, Macrostigma, and Tripetalum. [Sweeney, P. W. (2008). Phylogeny and Floral Diversity in the Genus Garcinia (Clusiaceae) and Relatives. International Journal of Plant Sciences, 169(9), 1288–1303.]

2) The lectotypification of Xanthochymus dulcis is not compelling and needs more work. See Forman (1997) for detailed information about the typification of Roxburgh names. Locating original material (i.e., specimens that are annotated by Roxburgh or are considered to have been associated by him to names in question) is often difficult. Roxburgh's protologue of X. dulcis implies that he based his discription on a tree living in the Botanical Garden at Calcutta before he departed India in 1813. No herbarium specimens were cited. Is there evidence that the purported syntypes were viewed/annotated by Roxburgh or somehow associated by him to the name in question? Following advice in Forman (1997), in addition to relevant specimens Roxburgh's illustration (Plate 270) in Plants of the Coast of Coromandel should be considered as a possible lectotype (see here: https://www.biodiversitylibrary.org/page/280299#page/119). Also, see additional comments in mark-up of version of ms provided with this review. [Forman, L.L., 1997. Notes concerning the typification of names of William Roxburgh's species of phanerogams. Kew Bulletin, pp.513-534.]

3) The report of the presence in G. prainiana of a disk/ring at the base of the pistillode is very surprising. To support the report of such an interesting result, the authors should include an image of a flower (from fresh or re-hydrated, dried material) that clearly shows this feature/structure. As stated previously, many morphological features and molecular data do not support the placement of G. prainiana into Section Xanthochymus, but rather support its placement into a clade that includes members of Jones' (1980) sections Macrostigma, Mungotia, and Triphyllum (part of Lineage B in Sweeney 2008). Species in these sections do not have "disks" or "rings" like those in Sections Xanthochyms, Rheedia, Teracentrum, or Rheediopsis (Lineage A in Sweeney 2008, see also Sweeney 2010). [Sweeney, P.W., 2008. Phylogeny and floral diversity in the genus Garcinia (Clusiaceae) and relatives. International Journal of Plant Sciences, 169(9), pp.1288-1303.]

4) The authors report that the pulp that surrounds the seeds is sarcotesta. A very important paper and interpretation has been overlooked by the authors. E. J. H. Corner studied the anatomy and morphology of Garcinia (and other Clusiaceae) fruits, publishing the results in The Seeds of Dicotyledons (1976, citation below). He interpreted the pulpy material that surrounds the seeds of species of Garcinia as endocarp. The testa was described as exarilate and unspecialized, except for secretory canals. [Corner, E.J.H., 1976. The Seeds of Dicotyledons: Volume 1 (Vols. 1 & 2). Cambridge University Press.

5) G. dulcis and G. xanthochymus are very morphologically similar and closely related and many previous authors have had difficulty distinguishing between these two species. The authors should consider adding a sentence or two that the distinctiveness of these two species is tentative and that detailed morphological (with statistical analyses) and molecular studies of these two species across their entire geographic range are needed. There are some additional papers that should be discussed and cited in this section of the manuscript about G. dulcis and G. xanthochymus: Maheshweri (1864) notes that G. xanthochyms has ciliate sepal tips and that G. dulcis does not. Can the authors comment on this character? Utami and Sari (2009) discuss the taxonomic status and distinguishing characteristics of G. duclis and G. xanthocyhymus. [Utami, N. and Sari, R., 2009. MUNDU: Garcinia xanthochymus Hook. f. ATAU G. dulcis (Roxb.) Kurz. Berita Biologi, 9(6), pp.739-744.]

Additional comments

The annotated version of the the manuscript that was uploaded with this review provides numerous minor comments and suggested edits and notes other issues with the manuscript that should be addressed/considered. View the annotated pdf with Adobe Acrobat.

Annotated reviews are not available for download in order to protect the identity of reviewers who chose to remain anonymous.

---

## Round 0.2 · Minor Revisions

Please take into account the few suggestions of Reviewer 2 indicated in the attached pdf. In my opinion, the white and black distribution maps contrast enormously with the images of living plants and the specimens. I recommend that you reconsider using color maps.

Reviewer 2 ·

Basic reporting

The authors have responded appropriately to my previous comments, and the manuscript has been improved as a result. I believe that the manuscript can be accepted as is.

Experimental design

No comment

Validity of the findings

No comment

Additional comments

No comment

Reviewer 3 ·

Basic reporting

Manuscript is much improved. A few minor things need to be addressed – see marked-up pdf for details.

Experimental design

No comment

Validity of the findings

No comment

Annotated reviews are not available for download in order to protect the identity of reviewers who chose to remain anonymous.

---

## Round 0.3 · accepted · Accept

I appreciate that you considered every issue raised by the reviewers in R2, they improved the manuscript. It is OK if you keep the black-and-white distribution maps.